# Adaptive Risk Minimization:
# Learning to Adapt to Domain Shift

**Marvin Zhang**[*1], **Henrik Marklund**[*2], **Nikita Dhawan**[*1],
**Abhishek Gupta**[1], **Sergey Levine**[1], **Chelsea Finn**[2]

[1] UC Berkeley, [2] Stanford University

## Abstract

A fundamental assumption of most machine learning algorithms is that the training and test data are drawn from the same underlying distribution. However, this assumption is violated in almost all practical applications: machine learning systems are regularly tested under *distribution shift*, due to changing temporal correlations, atypical end users, or other factors. In this work, we consider the problem setting of domain generalization, where the training data are structured into domains and there may be multiple test time shifts, corresponding to new domains or domain distributions. Most prior methods aim to learn a single robust model or invariant feature space that performs well on all domains. In contrast, we aim to learn models that *adapt* at test time to domain shift using unlabeled test points. Our primary contribution is to introduce the framework of adaptive risk minimization (ARM), in which models are directly optimized for effective adaptation to shift by learning to adapt on the training domains. Compared to prior methods for robustness, invariance, and adaptation, ARM methods provide performance gains of 1-4% test accuracy on a number of image classification problems exhibiting domain shift.

## 1 Introduction

The standard assumption in empirical risk minimization (ERM) is that the data distribution at test time will match the training distribution. When this assumption does not hold, i.e., when there is *distribution shift*, the performance of standard ERM methods can deteriorate significantly [54, 38].

As an example which we study quantitatively in Section 5, consider a handwriting classification model that, after training on data from past users, is deployed to new end users. Each new user represents a new test distribution that differs from the training distribution. Thus, each test setting involves dealing with shift. In Figure 1, we visualize a batch of 50 examples from a test user, and we highlight an ambiguous example which may be either a "2" (written with a loop) or an "a" (in the double-storey style) depending on the user's handwriting. Due to the biases in the training data, an ERM trained model incorrectly classifies this example as "2". However, we can see that the batch of images from this test user contains other examples of "2" (written without loops) and "a" (also double-storey) from this user. Can we somehow leverage this unlabeled data to better handle test shifts caused by new users?

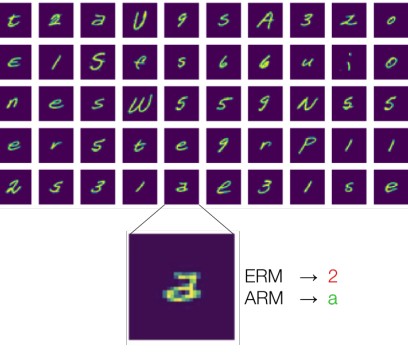

Figure 1: An example of ambiguous data points in handwriting classification, evaluated quantitatively in Section 5.

---

[*]equal contribution

35th Conference on Neural Information Processing Systems (NeurIPS 2021).

Any framework that aims to address this question must use additional assumptions beyond the ERM setting, and many such frameworks have been proposed [54]. One commonly used assumption within several frameworks, such as domain generalization [7, 23], is that the training data are provided in *domains* and distributions at test time will represent new domains. The example above neatly fits this description if we equate users with domains – we would be assuming that the training data are organized by users and that the model will be tested separately on new users, and these are reasonable assumptions. Constructing training domains in practice is generally accomplished by using meta-data, which exists for many commonly used datasets. Thus, this domain assumption is applicable for a wide range of realistic distribution shift problems (see, e.g., Koh et al. [35]).

However, prior benchmarks for domain generalization and similar settings typically center around *invariances* – i.e., in these benchmarks, there is a consistent input-output relationship across all domains, and the goal is to learn this relationship while ignoring the spurious correlations within the domains (see, e.g., Gulrajani and Lopez-Paz [23]). Thus, prior methods aim for generalization to shifts by discovering this relationship, through techniques such as robust optimization and learning an invariant feature space [41, 3, 60]. These methods are appealing in that they make minimal assumptions about the information provided at test time – in particular, they do not require test labels, and the learned model can be immediately applied to predict on a single point. Nevertheless, these methods also have limitations, such as in dealing with problems where the input-output relationship varies across domains, e.g., the handwriting classification example above.

In this paper, we instead focus on methods that aim to *adapt* at test time to domain shift. To do so, we study problems in which it is both feasible and helpful (and perhaps even necessary) to assume access to a batch or stream of inputs at test time. Leveraging this test assumption does not require labels for any test data and is feasible in many practical setups. For example, for handwriting classification, we do not access only single handwritten characters from an end user, but rather collections of characters such as sentences or paragraphs. Unlabeled adaptation has been shown empirically to be useful for distribution shift problems [69, 63, 75], such as for dealing with image corruptions [25]. Taking inspiration from these findings, we propose and evaluate on a number of problems, detailed in Section 5, for which adaptation is beneficial in dealing with domain shift.

Our main contribution is to introduce the framework of adaptive risk minimization (ARM), which proposes the following objective: optimize the model such that it can maximally leverage the unlabeled adaptation phase to handle domain shift. To do so, we instantiate a set of methods that, given a set of training domains, *meta-learns* a model that is adaptable to these domains. These methods are straightforward extensions of existing meta-learning approaches, thereby demonstrating that tools from the meta-learning toolkit can be readily adapted to tackle domain shift. Our experiments in Section 5 test on several image classification problems, derived from benchmarks for federated learning [9] and image classifier robustness [25], in which training and test domains share structure that can be leveraged for improved performance. These testbeds are also a contribution of our work, as we believe these problems can supplement existing benchmarks which, almost exclusively, are designed with invariance in mind [3, 53, 23]. We also evaluate on the WILDS suite of distribution shift problems [35], which have been curated to faithfully represent important real world problems. Empirically, we demonstrate that the proposed ARM methods, by leveraging meta-training and test time adaptation, are often able to outperform prior state-of-the-art methods by 1-4% test accuracy.

## 2 Related Work

A number of prior works have studied distribution shift in various forms [54]. In this section, we review prior work in domain generalization, group robustness, meta-learning, and adaptation.

**Invariance and robustness to domains.** As discussed above, a number of frameworks leverage training domains to address test time shift. The terminology in prior work is scattered and, depending on the application, includes terms such as "groups", "datasets", "subpopulations", and "users"; in this work, we adopt the term "domains" which we believe is an appropriate unifying term. A number of testbeds for this problem setting have been proposed for image classification, including generalizing to new datasets [17], new image types [39, 53], and underrepresented demographics [60].

Prior benchmarks typically assume the existence of a consistent input-output relationship across domains that is learnable by the specified model, thus motivating methods such as learning an invariant feature space [41, 44, 3] or optimizing for worst case group performance [30, 60]. In

particular, methods for domain generalization – sometimes referred to as multi-source domain adaptation [68] or zero shot domain adaptation [78] – have largely focused on learning invariant features [19, 67, 41, 44, 53]. Gulrajani and Lopez-Paz [23] provide a comprehensive survey of domain generalization benchmarks and find that, surprisingly, ERM is competitive with the state of the art across all the benchmarks considered. In Appendix C, we discuss this finding as well as the performance of an ARM method on this benchmark suite. In Section 5, we identify different problems for which adaptation is helpful, and we find that, on these problems, ARM methods consistently outperform ERM and other non adaptive methods for robustness and invariance.

**Meta-learning.** Meta-learning [62, 6, 71, 27] has been most extensively studied in the context of few shot *labeled* adaptation [61, 74, 55, 18, 65]. Our aim is not to address few-shot recognition problems, nor to propose a novel meta-learning algorithm, but rather to extend meta-learning paradigms to problems requiring unlabeled adaptation, with the primary goal of tackling distribution shift. This aim differs from previous work in meta-learning for domain generalization [40, 15], which seek to meta-train models for non adaptive generalization performance. We discuss in Section 4 how paradigms such as contextual meta-learning [20, 57] can be readily extended using the ARM framework.

Some other meta-learning methods adapt using both labeled and unlabeled data, either in the semi supervised learning setting [56, 81, 42] or the transductive learning setting [51, 46, 2, 29]. These works all assume access to labeled data for adaptation, whereas we propose methods and problems for purely unlabeled adaptation. Prior works in meta-learning for unlabeled adaptation include Yu et al. [80], who adapt a policy to imitate human demonstrations in the context of robotic learning; Metz et al. [48], who meta-learn an update rule for unsupervised representation learning, though they still require labels to learn a predictive model; and Alet et al. [1], who meta-learn adaptive models based on task specific unsupervised objectives. Unlike these prior works, we propose a general framework for tackling distribution shift problems by meta-learning unsupervised adaptation strategies. This framework simplifies the extension of meta-learning paradigms to these problems, encapsulates previous approaches such as the gradient based meta-learning approach of Yu et al. [80], and sheds light on how to improve existing strategies such as adaptation via batch normalization [43].

**Adaptation to shift.** Unlabeled adaptation has primarily been studied separately from meta-learning. Domain adaptation is a prominent framework that assumes access to test examples at *training* time [13, 76], similar to transductive learning [73]. As such, most domain adaptation methods consider the problem setting where there is a single test distribution [64, 14, 22, 19, 72, 10], and some of these methods are difficult to apply to problems where there are multiple test distributions. Certain domain adaptation methods have also been applied in the domain generalization setting, such as methods for learning invariant features [19, 67, 41], and we compare to these methods in Section 5.

Adaptive methods for domain generalization include Muandet et al. [49] and Kumagai and Iwata [37], who propose a method similar to one of the ARM methods described below. We compare to a version of this method in Appendix E. Blanchard et al. [7] and Blanchard et al. [8] provide a theoretic study of domain generalization and establish favorable generalization bounds for models that can adapt to domain shift at test time. We summarize some of these results in Section 3. In comparison, our work establishes a framework that makes explicit the connection between adaptation to domain shift and meta-learning, allowing us to devise new methods in a straightforward and principled manner. These methods are amenable to expressive models such as deep neural networks, which enables us to propose and evaluate on problems with raw image observations.

Test time adaptation has also been studied for dealing with label shift [59, 45, 66] and crafting favorable inductive biases for the domain of interest. For image classification, techniques such as normalizing via the test inputs [43] and optimizing self-supervised surrogate losses [69] have proven effective for adapting to image corruptions [25]. We compare to these prior methods in Section 5 and empirically demonstrate the advantage of using training domains to learn how to adapt.

## 3  Preliminaries and Notation

In this section, we discuss the domain generalization problem setting and formally describe adaptive models. In Section 4, we discuss how adaptive models can be meta-trained via the ARM objective and approach, and we instantiate ARM methods which we empirically evaluate in Section 5.

Let $\mathbf{x} \in \mathcal{X}$ and $y \in \mathcal{Y}$ represent the input and output, respectively. We can formalize the domain generalization problem setting using the following data generation process [8]: first, a joint data distribution $p_{\mathbf{x}y}$ is sampled from a set of distributions $\mathcal{P}_{\mathbf{x}y}$, and then some data points are sampled from $p_{\mathbf{x}y}$.[1] We refer to each $p_{\mathbf{x}y}$ as a domain, e.g., a particular dataset or user, thus $\mathcal{P}_{\mathbf{x}y}$ represents the set of all possible domains. We assume that the training dataset is composed of data from $S$ runs of this generative process, organized by domain. An equivalent characterization which we will use for clarity is that, within the training set, there are $S$ domains, and each data point $(\mathbf{x}^{(i)}, y^{(i)})$ is annotated with a domain label $z^{(i)}$. Each $z^{(i)}$ is an integer that takes on a value between 1 and $S$, indicating which $p_{\mathbf{x}y}$ generated the $i$-th training point (though, of course, we do not have access to, or knowledge of, $p_{\mathbf{x}y}$ itself). At test time, there may be multiple evaluation settings, where each setting is considered separately and contains only *unlabeled* data sampled via a new run of the same generative process. This data may represent, e.g., a new dataset or user, and the test domains are likely to be distinct from the training domains when $|\mathcal{P}_{\mathbf{x}y}|$ is large or infinite.

Our formal goal is to optimize for expected performance, e.g., classification accuracy, at test time. To do so, let us first consider predictive models of the form $f : \mathcal{X} \times \mathcal{P}_{\mathbf{x}} \to \mathcal{Y}$, where the model $f$ takes in not just an input $\mathbf{x}$ but also the marginal input distribution $p_{\mathbf{x}} \in \mathcal{P}_{\mathbf{x}}$ that $\mathbf{x}$ was sampled from. We refer to $f$ as an *adaptive* model, as it has the opportunity to use $p_{\mathbf{x}}$ to adapt its predictions on $\mathbf{x}$. The underlying assumption is that $p_{\mathbf{x}}$ provides information about $p_{y|\mathbf{x}}$, i.e., $p_{\mathbf{x}}$ is used as a surrogate input in place of $p_{\mathbf{x}y}$. In the worst case, if $p_{\mathbf{x}}$ and $p_{y|\mathbf{x}}$ are sampled independently, then the model does not benefit at all from knowing $p_{\mathbf{x}}$. In many problems, however, we expect knowledge about $p_{\mathbf{x}}$ to be useful, e.g., for resolving ambiguity as in the handwriting classification example in Section 1.

Theoretically, when $p_{\mathbf{x}}$ provides information about $p_{y|\mathbf{x}}$, and when training and test domains are drawn from the same distribution over $\mathcal{P}_{\mathbf{x}y}$, we can establish favorable generalization bounds for the expected performance of $f$ in adapting to domain shift at test time. We can formalize this as follows. First, define a *prediction model* to be a non adaptive model of the form $g : \mathcal{X} \to \mathcal{Y}$, and define the risk for a prediction model $g$ and loss function $\ell$, under a data distribution $p_{\mathbf{x}y}$, as

$$\mathcal{R}(g, p_{\mathbf{x}y}) \triangleq \mathbb{E}_{p_{\mathbf{x}y}} \left[ \ell(g(\mathbf{x}), y) \right] .$$

Further, define the Bayes optimal risk for $\ell$ under $p_{\mathbf{x}y}$ as

$$\mathcal{R}^{\star}(p_{\mathbf{x}y}) \triangleq \min_g \mathcal{R}(g, p_{\mathbf{x}y}) .$$

Let $\mu$ denote the distribution on $\mathcal{P}_{\mathbf{x}y}$ from which training and test domains $p_{\mathbf{x}y}$ are sampled. To avoid overlapping terms, define the *adaptive risk* for an adaptive model $f$ and $\ell$, under $\mu$, to be

$$\mathcal{E}(f, \mu) \triangleq \mathbb{E}_\mu \left[ \mathbb{E}_{p_{\mathbf{x}y}} \left[ \ell(f(\mathbf{x}, p_{\mathbf{x}}), y) \right] \right] . \tag{1}$$

We state the following result from Blanchard et al. [8], which details a condition on $\mu$ under which $\mathcal{E}$ is a strongly principled objective for learning adaptive models.

**Lemma 9 from Blanchard et al. [8].** Let $f^{\star}$ denote a minimizer of $\mathcal{E}$ for the given $\mu$. If $\mu$ is a distribution on $\mathcal{P}_{\mathbf{x}y}$ such that $\mu$-almost surely it holds that $p_{y|\mathbf{x}} = M(p_{\mathbf{x}})$ for some deterministic mapping $M$, then for $\mu$-almost all $p_{\mathbf{x}y}$, we have

$$\mathcal{R}(f^{\star}(\cdot, p_{\mathbf{x}}), p_{\mathbf{x}y}) = \mathcal{R}^{\star}(p_{\mathbf{x}y}) \implies \mathcal{E}(f^{\star}, \mu) = \mathbb{E}_\mu \left[ \mathcal{R}^{\star}(p_{\mathbf{x}y}) \right] .$$

In other words, an adaptive model which minimizes the adaptive risk $\mathcal{E}$ coincides with a Bayes optimal decision function for $p_{\mathbf{x}y}$, *for $\mu$-almost all domains $p_{\mathbf{x}y}$.*

**Remark.** The required condition on $\mu$ – that $p_{y|\mathbf{x}}$ is determined by $p_{\mathbf{x}}$ – holds if, and only if, an expert (or oracle) is able to correctly label inputs from a given domain provided only information about the input distribution. This condition holds for the testbeds proposed in this paper, those in Gulrajani and Lopez-Paz [23], and those in WILDS [35]. The condition does not hold for, e.g., standard few shot learning testbeds, where it is possible for two domains with identical input distributions to shuffle their label orderings differently [74]. Thus, these problems are outside the scope of this work.

This result provides strong justification for learning adaptive models $f$ by minimizing the adaptive risk $\mathcal{E}$. However, a practical instantiation of this approach requires some approximations. First, we

---

[1]Formally, the number of points sampled is another random variable with support over the positive integers.

do not know and cannot input $p_\mathbf{x}$ to $f$ in most cases. Instead, we instantiate $f$ such that it takes in a batch of inputs $\mathbf{x}_1, \ldots, \mathbf{x}_K$, all from the same domain, where $K$ can vary. $f$ makes predictions on the whole batch, which also serves as an empirical approximation (i.e., a histogram) $\hat{p}_\mathbf{x}$ of $p_\mathbf{x}$ [8]. In our exposition, we will assume that a batch of unlabeled points is available at test time for adaptation. However, we also experiment in Section 5 with the *streaming* setting where the test inputs are observed one at a time and adaptation occurs incrementally.

Notice that, if we instead passed in an approximation $\hat{p}_{\mathbf{x}y}$ of $p_{\mathbf{x}y}$ to the model, such as a batch of *labeled* data $(\mathbf{x}_1, y_1), \ldots, (\mathbf{x}_K, y_K)$, then this setup would resemble the standard few shot meta-learning problem [74]. Formally, a meta-learning model takes in both an input $\mathbf{x}$ and $\hat{p}_{\mathbf{x}y}$, which approximates the distribution that $\mathbf{x}$ was sampled from and thus can be used to adapt the prediction on $\mathbf{x}$. Compared to our problem setting, the meta-learning formalism can tackle a wider range of problems but also requires more restrictive assumptions, specifically, labels at test time via $\hat{p}_{\mathbf{x}y}$. *Transductive meta-learning* methods further assume that, in addition to $\hat{p}_{\mathbf{x}y}$, a full batch of inputs $\mathbf{x}_1, \ldots, \mathbf{x}_K$ is passed into the model, which allows for better estimation of the input distribution $p_\mathbf{x}$ [51, 46, 29]. The model then makes predictions on this entire batch. In meta-learning terminology, $\hat{p}_{\mathbf{x}y}$ and $\mathbf{x}_1, \ldots, \mathbf{x}_K$ are often referred to as the *support* and *query*, respectively. Therefore, another interpretation of the adaptive models that we study in this work is that they resemble transductive meta-learning models, but they are given only the unlabeled query and not the labeled support set.

In the next section, we expand on this connection to develop the ARM framework, which then allows us to bring forward tools from meta-learning to tackle domain shift problems.

## 4    Adaptive Risk Minimization

In this section, we formally describe the ARM framework, which defines an objective for training adaptive models to tackle domain shift. Furthermore, we propose a general meta-learning algorithm as well as specific methods for optimizing the ARM objective. In Section 5, we test these ARM methods on problems for which unlabeled adaptation can be leveraged for better test performance.

### 4.1    Devising the ARM objective

We wish to learn an adaptive model $f : \mathcal{X}^K \to \mathcal{Y}^K$ to tackle domain shift. As noted, meta-learning methods for labeled adaptation study a similar form of model, and a common approach in many of these methods is to define $f$ such that it is composed of two parts: first, a *learner* which ingests the data and produces parameters, and second, a *prediction model* which uses these parameters to make predictions [74, 18]. We will follow a similar strategy which, as we will discuss in subsection 4.2, allows us to easily extend and design meta-learning methods towards our goal.

In particular, we will decompose the model $f$ into two modules: a standard prediction model $g(\cdot\,; \theta) : \mathcal{X} \to \mathcal{Y}$, that is parameterized by $\theta \in \Theta$ and predicts $y$ given $\mathbf{x}$, and an *adaptation model* $h(\cdot\,, \cdot\,; \phi) : \Theta \times \mathcal{X}^K \to \Theta$, which is parameterized by $\phi$. $h$ takes in the prediction model parameters $\theta$ and $K$ unlabeled data points and uses the $K$ points to produce adapted parameters $\theta'$. This is analogous to the learner in meta-learning, however, $h$ adapts the model parameters using only unlabeled data. We defer the discussion of how to instantiate $h$ to subsection 4.2.[2]

The ARM objective is to optimize $\phi$ and $\theta$ such that $h$ can adapt $g$ using unlabeled data sampled according to a particular domain $z$. This can be expressed as the optimization problem

$$\min_{\theta, \phi} \hat{\mathcal{E}}(\theta, \phi) = \mathbb{E}_{p_z}\left[ \mathbb{E}_{p_{\mathbf{x}y|z}} \left[ \frac{1}{K} \sum_{k=1}^{K} \ell(g(\mathbf{x}_k; \theta'), y_k) \right] \right] , \text{ where } \theta' = h(\theta, \mathbf{x}_1, \ldots, \mathbf{x}_K; \phi) . \quad (2)$$

Note that $\hat{\mathcal{E}}$ is the empirical form of the adaptive risk in Equation 1 for the form of $f$ we have defined. Mimicking the generative process from Section 3 that we assume generated the training data, $p_z$ is a categorical distribution over $\{1, \ldots, S\}$ which places uniform probability mass on each training domain, and $p_{\mathbf{x}y|z}$ assigns uniform probability to only the training points within a particular domain. As we have established theoretically, we expect the trained models to perform well at test time if the

---

[2]For some meta-learning methods, the learner does not take as input the unadapted model parameters [74], and we also devise some methods of this form. In the formalism above, these methods simply ignore the input $\theta$.

test domains are sampled independently and identically – i.e., from the same distribution over $\mathcal{P}_{\mathbf{x}y}$ – as the training domains. In practice, similar to how meta-learned few shot classification models are evaluated on new and unseen meta-test classes [74, 18], we empirically show in Section 5 that the trained models can generalize to test domains that are not sampled identically to the training domains.

## 4.2 Optimizing the ARM objective

Algorithm 1 presents a general meta-learning approach for optimizing the ARM objective. As described above, $h$ outputs updated parameters $\theta'$ using an unlabeled batch of data (line 5). This mimics the adaptation procedure at test time, where we do not assume access to labels (lines 7-8). However, the training update itself does rely on the labels (line 6). We assume that $h$ is differentiable with respect to its input $\theta$ and $\phi$, thus we use gradient updates on both $\theta$ and $\phi$ to optimize for *post adaptation* performance on a mini batch of data sampled according to a particular domain $z$. In practice, we also sample mini batches of domains, rather than just one domain (as written in line 3), to provide a better gradient signal for optimizing $\phi$ and $\theta$.

---

**Algorithm 1** Meta-Learning for ARM

```
// Training procedure
```
**Require:** # training steps $T$, batch size $K$, learning rate $\eta$
1: **Initialize:** $\theta, \phi$
2: **for** $t = 1, \ldots, T$ **do**
3:    Sample $z$ uniformly from training domains
4:    Sample $(\mathbf{x}_k, y_k) \sim p(\cdot, \cdot | z)$ for $k = 1, \ldots, K$
5:    $\theta' \leftarrow h(\theta, \mathbf{x}_1, \ldots, \mathbf{x}_K; \phi)$
6:    $(\theta, \phi) \leftarrow (\theta, \phi) - \eta \nabla_{(\theta, \phi)} \sum_{k=1}^{K} \ell(g(\mathbf{x}_k; \theta'), y_k)$

```
// Test time adaptation
procedure
```
**Require:** $\theta, \phi$, test batch $\mathbf{x}_1, \ldots, \mathbf{x}_K$
7: $\theta' \leftarrow h(\theta, \mathbf{x}_1, \ldots, \mathbf{x}_K; \phi)$
8: $\hat{y}_k \leftarrow g(\mathbf{x}_k; \theta')$ for $k = 1, \ldots, K$

---

Together, Equation 2 and Algorithm 1 shed light on a number of ways to devise methods for solving the ARM problem. First, we can extend meta-learning paradigms to the ARM problem setting, and any paradigm in which the adaptation model $h$ can be augmented to operate on unlabeled data is readily applicable. As an example, we propose the ARM-CML method, which is inspired by recent works in contextual meta-learning (CML) [20, 57]. Second, we can enhance prior unlabeled adaptation methods by incorporating a meta-training phase that allows the model to better leverage the adaptation. To this end, we propose the ARM-BN method, based on the general approach of adapting using batch normalization (BN) statistics of the test inputs [43, 63, 33, 50]. Third, we can incorporate existing methods for meta-learning unlabeled adaptation to solve domain shift problems. We demonstrate this by proposing the ARM-LL method, which is based on the robotic imitation learning method from Yu et al. [80] which adapts via a learned loss (LL). All of these methods are straightforward extensions of existing meta-learning and adaptation methods, and this is intentional – we aim to show how existing tools can be readily adapted to tackle domain generalization problems. We summarize the methods here and refer the reader to Appendix B for complete details.

**ARM-CML.** In ARM-CML, the parameters $\phi$ of $h$ define the weights of a *context network* $f_{\text{cont}}(\cdot; \phi) : \mathcal{X} \rightarrow \mathbb{R}^D$, parameterized by the adaptation model parameters $\phi$. We also instantiate the model with a *prediction network* $f_{\text{pred}}(\cdot, \cdot; \theta) : \mathcal{X} \times \mathbb{R}^D \rightarrow \mathcal{Y}$, parameterized by $\theta$. When given a mini batch of inputs, $f_{\text{cont}}$ processes each example $\mathbf{x}_k$ in the mini batch separately and outputs $\mathbf{c}_k \in \mathbb{R}^D$ for $k = 1, \ldots, K$, which are averaged together into a *context* $\mathbf{c} = \frac{1}{K} \sum_{k=1}^{K} \mathbf{c}_k$. $D$ is a hyperparameter, and in our experiments, we choose $D$ to be the dimensionality of $\mathbf{x}$, such that we can concatenate each image $\mathbf{x}_k$ and the context $\mathbf{c}$ along the channel dimension to produce the input to $f_{\text{pred}}$. In other words, $f_{\text{pred}}$ processes each $\mathbf{x}_k$ separately to produce an estimate of the output $\hat{y}_k$, but it additionally receives $\mathbf{c}$ as input. In this way, $f_{\text{cont}}$ can provide information about the entire batch of $K$ unlabeled data points to $f_{\text{pred}}$ for predicting the correct outputs.

Note that the difference between ARM-CML and prior contextual meta-learning approaches is that, in prior approaches, the context network processes both inputs and outputs to produce each $\mathbf{c}_k$. ARM-CML is designed for the domain generalization setting in which we do not assume access to labels at test time, thus we meta-train for unlabeled adaptation performance at training time.

**ARM-BN.** ARM-BN is a particularly simple method that is applicable for any model $g$ that has BN layers [31]. Practically, training $g$ via ARM-BN follows the same protocol as Ioffe and Szegedy [31] except for two key differences: first, the training batches are sampled from a single domain, rather than from the entire dataset, and second, the normalization statistics are recomputed at test time rather than using a training running average. As noted, this second difference has been explored

by several works as a method for test time adaptation, but the first difference is novel to ARM-BN. Following Algorithm 1, ARM-BN defines a meta-training procedure in which $g$ learns to adapt – i.e., compute normalization statistics – using batches of training points sampled from the same domain. We empirically show in Section 5 that, for problems where BN adaptation already has a favorable inductive bias, such as for image classification, further introducing meta-training boosts its performance. We believe that other test time adaptation methods, such as those based on optimizing surrogate losses [69, 75], may similarly benefit from their corresponding meta-training procedures.

At a high level, ARM-BN operates in a similar fashion to ARM-CML, thus we group these methods together into the umbrella of contextual approaches, shown in Figure 2 (top). The interpretation of ARM-BN through the contextual approach is that $h$ replaces the running statistics used by standard BN with statistics computed on the batch of inputs, which then serves as the context $\mathbf{c}$. Thus, for ARM-BN, there is no context network, and $h$ has no parameters beyond the model parameters $\theta$ involved in computing BN statistics. The model $g$ is again specified via a prediction network $f_{pred}$, which must have BN layers. BN typically tracks a running average of the first and second moments of the activations in these layers, which are then used at test time. ARM-BN defines $h$ such that it swaps out these moments for the moments computed via the activations on the test batch, thus giving us adapted parameters $\theta'$ if we view the moments as part of the model parameters. This method is remarkably simple, and in deep learning libraries such as PyTorch [52], implementing ARM-BN involves changing a single line of code. However, as shown in Section 5, this method also performs very well empirically, and the adaptation effectiveness is further boosted by meta-training.

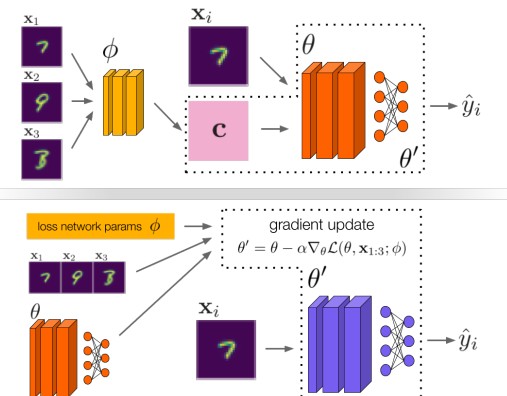

Figure 2: In the contextual approach (top), $\mathbf{x}_1, \ldots, \mathbf{x}_K$ are summarized into a context $\mathbf{c}$, and we propose two methods for this summarization, either through a separate context network or using batch normalization activations in the model itself. $\mathbf{c}$ can then be used by the model to infer additional information about the input distribution. In the gradient based approach (bottom), an unlabeled loss function $\mathcal{L}$ is used for gradient updates to the model parameters, in order to produce parameters that are specialized to the test inputs and can produce more accurate predictions.

**ARM-LL.** ARM-LL, depicted in Figure 2 (bottom), follows the gradient based meta-learning paradigm [18] and learns parameters $\theta$ that are amenable to gradient updates on a loss function in order to quickly adapt to a new problem. In other words, $h$ produces $\theta' = \theta - \alpha \nabla_\theta \mathcal{L}(\theta, \mathbf{x}_1, \ldots, \mathbf{x}_K; \phi)$, where $\alpha$ is a hyperparameter. Note that the loss function $\mathcal{L}$ used in the gradient updates is different from the original supervised loss function $\ell$, in that it operates on only the inputs $\mathbf{x}$, rather than the input output pairs that $\ell$ receives. We follow the general implementation of this approach proposed in Yu et al. [80]. We define $g$ to produce output features $\mathbf{o} \in \mathbb{R}^{|\mathcal{Y}|}$ that are used as logits when making predictions. We then define the unlabeled loss function $\mathcal{L}$ to be the composition of $g$ and a *loss network* $f_{loss}(\cdot; \phi) : \mathbb{R}^{|\mathcal{Y}|} \to \mathbb{R}$, which takes in the output features from $g$ and produces a scalar. We use the $\ell_2$-norm of these scalars across the batch of inputs as the loss for updating $\theta$. In other words,

$$h(\theta, \mathbf{x}_1, \ldots, \mathbf{x}_K; \phi) = \theta - \alpha \nabla_\theta \|\mathbf{v}\|_2, \text{ where } \mathbf{v} = [f_{loss}(g(\mathbf{x}_1; \theta); \phi), \ldots, f_{loss}(g(\mathbf{x}_K; \theta); \phi)].$$

## 5 Experiments

Our experiments are designed to answer the following questions:

1. Do ARM methods learn models that can leverage unlabeled adaptation to tackle domain shift?
2. How do ARM methods compare to prior methods for robustness, invariance, and adaptation?
3. Can models trained via ARM methods adapt successfully in the streaming test setting?

### 5.1 Evaluation domains and protocol

We propose four image classification problems, which we present below and describe in full detail in Appendix D. We also present results on datasets from the WILDS benchmark [35] in subsection 5.4.

We believe that the problems we propose in this paper can supplement existing benchmarks for domain shift, which, as discussed above, are designed to test invariances. A key characteristic of the problems presented here is the potential for adaptation to improve test performance, and this differs from prior benchmarks such as the problems compiled by DomainBed [23]. In Appendix C, we compare our testbeds to DomainBed and group robustness benchmarks, and we briefly discuss the results in Gulrajani and Lopez-Paz [23], which also evaluate ARM-CML.

**Rotated MNIST.** We study a modified version of MNIST where images are rotated in 10 degree increments, from 0 to 130 degrees. We treat each rotation as a separate domain, i.e., a different value of $z$. We use only 108 training data points for each of the 2 smallest domains (120 and 130 degrees), and 324 points each for rotations 90 to 110, whereas the overall training set contains 32292 points. In this setting, we hypothesize that adaptation can specialize the model to specific domains, in particular the *rare domains* in the training set. For each test evaluation, we generate images from the MNIST test set with a certain rotation. We measure both worst case and average accuracy across domains.

**Federated Extended MNIST (FEMNIST).** The extended MNIST (EMNIST) dataset consists of images of handwritten uppercase and lowercase letters, in addition to digits [12]. FEMNIST is the same dataset, but it also provides the meta-data of which user generated each data point [9]. We treat each user as a domain. We measure each method's worst case and average accuracy across 35 test users, which are held out and thus *disjoint* from the training users. As discussed in Section 1, adaptation may help for this problem for specializing the model and resolving ambiguous data points.

**Corrupted image datasets.** CIFAR-10-C and Tiny ImageNet-C [25] augment the CIFAR-10 [36] and Tiny ImageNet test sets with common image corruptions that vary in type and severity. The original goal of these augmented test sets was to benchmark how well methods could handle these corruptions without access to *any* corruptions during training [25]. Thus, successful methods for these problems typically have relied on domain knowledge and heuristics designed specifically for image classification. For example, prior work has shown that carefully designed test time adaptation procedures are effective for these problems [69, 63, 75]. One possible reason for this phenomenon is that convolutional networks are biased toward texture [21], which is distorted by corruptions, thus adaptation can help the model recover its performance for each corruption type.

We study whether meta-training for adaptation performance can improve upon these results. To do so, we modify the protocol from Hendrycks and Dietterich [25] to fit into the ARM problem setting by applying a set of 56 corruptions to the training data, and we define each corruption to be a domain. We use a *disjoint* set of 22 corruptions for the test data, which are mostly of different types from the training corruptions (thus, not sampled identically), and we measure worst case and average accuracy across the test corruptions. This modification allows us to study, for both ARM and prior methods, whether seeing corruptions at training time can help the model deal with new corruptions at test time.

## 5.2 Comparisons and ablations

We compare the ARM methods against several prior methods designed for robustness, invariance, and adaptation. We describe the comparisons here and provide additional details in Appendix D.

**Test time adaptation.** We evaluate the general approach of using test batches to compute BN statistics [43, 63, 33, 50], which we term BN adaptation. We also compare to test time training (TTT) [69], which adapts the model at test time using a self-supervised rotation prediction loss. These methods have previously achieved strong results for image classification, likely because they constitute favorable inductive biases for improving on the true classification task [69].

**Ablations.** We also include ablations of the ARM-CML and ARM-LL methods, which sample training batches of unlabeled examples uniformly from the entire training set, rather than sampling from a single domain.[3] These "context ablation" and "learned loss ablation" are similar to test time adaptation methods in that they do not require training domains, thus they allow us to evaluate whether or not meta-training on domain shifts is important for improved performance.

**Group robustness and invariance.** Sagawa et al. [60] recently proposed a state-of-the-art method for group robustness, and we refer to this approach as distributionally robust neural networks (DRNN). Their work also evaluates a strong upweighting (UW) baseline that samples uniformly from each group, and so we also evaluate this approach in our experiments. Additionally, we compare to

---

[3]Note that the corresponding ablation of ARM-BN is simply the BN adaptation method.

Table 1: Worst case (WC) and average (Avg) top 1 accuracy on all testbeds, where means and standard errors are reported across three separate runs of each method. Horizontal lines separate methods that make use of (from top to bottom): neither, training domains, test batches, or both. ARM methods consistently achieve greater robustness, measured by WC, and Avg performance compared to prior methods. *UW is identical to ERM for CIFAR-10-C and Tiny ImageNet-C.

These results have been updated from an earlier version of the paper, primarily for CIFAR-10-C, due to significant refactoring of the code, additional hyperparameter tuning for both the ARM methods and the prior methods, and efforts to standardize results across the authors' different computing environments and library versions. These results are reproducible from the publicly available code: `https://github.com/henrikmarklund/arm`.

| Method | MNIST | | FEMNIST | | CIFAR-10-C | | Tiny ImageNet-C | |
| --- | --- | --- | --- | --- | --- | --- | --- | --- |
| | WC | Avg | WC | Avg | WC | Avg | WC | Avg |
| ERM | $74.5 \pm 1.4$ | $93.6 \pm 0.4$ | $62.4 \pm 0.4$ | $79.1 \pm 0.3$ | $54.1 \pm 0.3$ | $70.4 \pm 0.1$ | $20.3 \pm 0.5$ | $41.9 \pm 0.1$ |
| UW* | $\mathbf{80.3 \pm 1.2}$ | $\mathbf{95.1 \pm 0.1}$ | $\mathbf{65.7 \pm 0.7}$ | $80.3 \pm 0.6$ | — | — | — | — |
| DRNN | $79.9 \pm 0.7$ | $\mathbf{94.9 \pm 0.1}$ | $57.5 \pm 1.7$ | $76.5 \pm 1.2$ | $49.3 \pm 0.9$ | $65.7 \pm 0.5$ | $14.2 \pm 0.2$ | $31.6 \pm 1.0$ |
| DANN | $78.8 \pm 0.8$ | $\mathbf{94.9 \pm 0.1}$ | $\mathbf{65.4 \pm 1.0}$ | $\mathbf{81.7 \pm 0.3}$ | $\mathbf{53.9 \pm 2.2}$ | $\mathbf{69.8 \pm 0.3}$ | $\mathbf{20.4 \pm 0.7}$ | $\mathbf{40.9 \pm 0.2}$ |
| MMD | $\mathbf{82.4 \pm 0.9}$ | $\mathbf{95.3 \pm 0.3}$ | $62.4 \pm 0.7$ | $79.8 \pm 0.4$ | $52.2 \pm 0.3$ | $69.5 \pm 0.1$ | $19.7 \pm 0.2$ | $40.1 \pm 0.1$ |
| BN adaptation | $78.0 \pm 0.3$ | $94.4 \pm 0.1$ | $65.7 \pm 1.5$ | $80.0 \pm 0.5$ | $60.6 \pm 0.3$ | $70.9 \pm 0.1$ | $26.5 \pm 0.3$ | $\mathbf{42.8 \pm 0.0}$ |
| TTT | $\mathbf{81.1 \pm 0.3}$ | $\mathbf{95.4 \pm 0.1}$ | $\mathbf{68.6 \pm 0.4}$ | $\mathbf{84.2 \pm 0.1}$ | $\mathbf{61.5 \pm 0.3}$ | $\mathbf{71.7 \pm 0.5}$ | $\mathbf{27.6 \pm 0.5}$ | $37.7 \pm 0.3$ |
| CML ablation | $63.5 \pm 1.8$ | $90.1 \pm 0.2$ | $61.8 \pm 0.8$ | $81.6 \pm 0.5$ | $58.8 \pm 0.1$ | $69.6 \pm 0.2$ | $26.3 \pm 0.6$ | $42.5 \pm 0.1$ |
| LL ablation | $79.9 \pm 1.1$ | $95.0 \pm 0.3$ | $64.1 \pm 1.6$ | $80.8 \pm 0.2$ | $\mathbf{60.9 \pm 0.4}$ | $71.3 \pm 0.0$ | $21.6 \pm 2.1$ | $32.6 \pm 3.2$ |
| ARM-CML | $88.0 \pm 0.8$ | $96.3 \pm 0.4$ | $\underline{\mathbf{70.9 \pm 1.4}}$ | $\underline{\mathbf{86.4 \pm 0.3}}$ | $61.2 \pm 0.4$ | $70.3 \pm 0.2$ | $\mathbf{29.1 \pm 0.4}$ | $\underline{\mathbf{43.3 \pm 0.1}}$ |
| ARM-BN | $83.3 \pm 0.5$ | $95.6 \pm 0.1$ | $64.5 \pm 3.2$ | $83.2 \pm 0.5$ | $\underline{\mathbf{61.7 \pm 0.3}}$ | $\mathbf{72.4 \pm 0.3}$ | $\underline{\mathbf{28.3 \pm 0.3}}$ | $\underline{\mathbf{43.3 \pm 0.1}}$ |
| ARM-LL | $\underline{\mathbf{88.9 \pm 0.8}}$ | $\underline{\mathbf{96.9 \pm 0.2}}$ | $67.0 \pm 0.9$ | $84.3 \pm 0.7$ | $\mathbf{61.2 \pm 0.7}$ | $\underline{\mathbf{72.5 \pm 0.4}}$ | $25.4 \pm 0.1$ | $35.7 \pm 0.4$ |

domain adversarial neural networks (DANN) [19] and maximum mean discrepancy (MMD) feature learning [41], two state-of-the-art methods for adversarial learning of invariant predictive features. For the WILDS datasets, we include the numbers reported in Koh et al. [35] for DRNN and two other invariance methods, correlation alignment (CORAL) [67] and invariant risk minimization (IRM) [3].

Robustness and invariance methods assume access to training domains but not test batches, whereas adaptation methods assume the opposite. Thus, at a high level, we can view the comparisons to these methods as evaluating the importance of each of these assumptions for the specified problems.

## 5.3  Quantitative evaluation and comparisons

The results for the four proposed benchmarks are presented in Table 1. The best results, stratified by classes of methods, are bolded, with the single best result across all methods underlined. Across all of these problems, ARM methods increase both worst case and average accuracy compared to all other methods. ARM-CML performs well across all tasks, and despite its simplicity, ARM-BN achieves the best performance overall on the corrupted image testbeds, demonstrating the effectiveness of meta-training on top of an already strong adaptation procedure. BN adaptation and TTT are the strongest prior methods, as these adaptation procedures constitute inductive biases that are generally well suited for image classification. However, ARM methods are comparatively less reliant on favorable inductive biases and consistently attain better results. In general, we observe poor performance from robustness methods, varying performance from invariance methods, strong performance from adaptation methods, and the strongest performance from ARM methods.

When we cannot access a batch of test points all at once, and instead the points are observed in a streaming fashion, we can augment the proposed ARM methods to perform sequential model updates. For example, ARM-CML and ARM-BN can update their average context and normalization statistics, respectively, after observing each new test point. In Figure 3, we study this test setting for the Tiny ImageNet-C problem. We see that both models trained with ARM-CML and ARM-BN are able to achieve near their original worst case and average accuracy within observing 50 data points, well before the training batch size of 100. This result demonstrates that ARM methods are applicable for problems where test points must be observed one at a time, provided that the model is permitted to adapt using each point. We describe in detail how each ARM method can be applied to the streaming setting in Appendix B, and we provide streaming results on rotated MNIST in Appendix E.

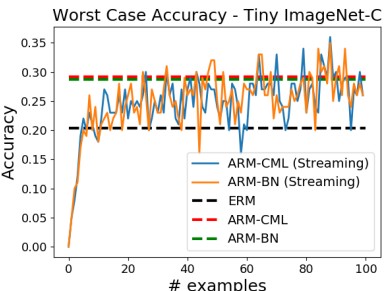 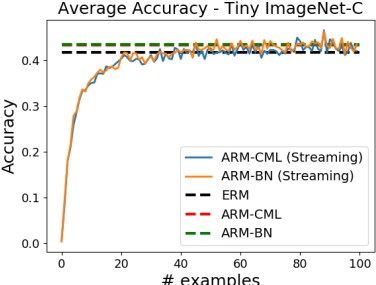

Figure 3: In the streaming setting, ARM methods reach strong performance on Tiny ImageNet-C after fewer than 50 data points, despite using training batch sizes of 100. This highlights that the trained models are able to adapt successfully in the standard streaming evaluation setting.

Table 2: Results on the WILDS image testbeds. Different methods are best suited for different problems, motivating the need for a wide range of methods. ARM-BN struggles on FMoW but performs well on the other datasets, in particular RxRx1.

| | iWildCam | | Camelyon17 | RxRx1 | FMoW | | PovertyMap | |
|---|---|---|---|---|---|---|---|---|
| Method | Acc | Macro F1 | Acc | Acc | WC Acc | Avg Acc | WC Pearson r | Pearson r |
| ERM | $71.6 \pm 2.5$ | $31.0 \pm 1.3$ | $70.3 \pm 6.4$ | $29.9 \pm 0.4$ | $32.3 \pm 1.25$ | $53.0 \pm 0.55$ | $0.45 \pm 0.06$ | $0.78 \pm 0.04$ |
| DRNN | $72.7 \pm 2.0$ | $23.9 \pm 2.1$ | $68.4 \pm 7.3$ | $23.0 \pm 0.3$ | $30.8 \pm 0.81$ | $52.1 \pm 0.5$ | $0.39 \pm 0.06$ | $0.75 \pm 0.07$ |
| CORAL | $73.3 \pm 4.3$ | $32.8 \pm 0.1$ | $59.5 \pm 7.7$ | $28.4 \pm 0.3$ | $31.7 \pm 1.24$ | $50.5 \pm 0.36$ | $0.44 \pm 0.06$ | $0.78 \pm 0.05$ |
| IRM | $59.8 \pm 3.7$ | $15.1 \pm 4.9$ | $64.2 \pm 8.1$ | $8.2 \pm 1.1$ | $30.0 \pm 1.37$ | $50.8 \pm 0.13$ | $0.43 \pm 0.07$ | $0.77 \pm 0.05$ |
| BN adaptation | $46.4 \pm 1.0$ | $13.8 \pm 0.3$ | $88.6 \pm 1.4$ | $20.0 \pm 0.2$ | $30.2 \pm 0.26$ | $51.6 \pm 0.16$ | $0.39 \pm 0.17$ | $0.82 \pm 0.06$ |
| ARM-BN | $70.3 \pm 2.4$ | $23.2 \pm 2.7$ | $87.2 \pm 0.9$ | $31.2 \pm 0.1$ | $24.6 \pm 0.04$ | $42.0 \pm 0.21$ | $0.49 \pm 0.21$ | $0.84 \pm 0.05$ |

## 5.4 WILDS results

Finally, we present results on the WILDS benchmark [35] in Table 2. We evaluate BN adaptation and ARM-BN on these testbeds. We see that, on these real world distribution shift problems, different methods perform well for different problems. CORAL, a method for invariance [67], performs best on the iWildCam animal classification problem [5], whereas no methods outperform ERM by a significant margin on the FMoW [11] or PovertyMap [79] satellite imagery problems. ARM-BN performs particularly poorly on the FMoW problem. However, it performs well on PovertyMap and significantly improves performance on the RxRx1 [70] problem of treatment classification from medical images. On the other medical imagery problem of Camelyon17 [4] tumor identification, adaptation in general boosts performance dramatically. These results indicate the need to consider a wide range of tools, including meta-learning and adaptation, for combating distribution shift.

## 6 Discussion and Future Work

We presented adaptive risk minimization (ARM), a framework and problem formulation for learning models that can adapt in the face of domain shift at test time using only a batch of unlabeled test examples. We devised an algorithm and instantiated a set of methods for optimizing the ARM objective that meta-learns models that are adaptable to different domains of training data. Empirically, we observed that ARM methods consistently improve performance in terms of both average and worst case metrics, as compared to a number of prior approaches for handling domain shift.

Though we provided contextual meta-learning as a concrete example, a number of other meta-learning paradigms would also be interesting to extend to the ARM setting. For example, few shot generative modeling objectives would be a natural fit for unlabeled adaptation [16, 26, 77]. Another exciting direction for future work is to explore the problem setting where domains are not provided at training time. As discussed in Appendix E, in this setting, we can instead construct domains via unsupervised learning techniques. Similar to Hsu et al. [28], one promising approach is to generate a diverse set of domains in order to learn generally effective adaptation strategies. Robustness and invariance methods cannot be used easily with multiple different groupings, learned or otherwise, as techniques such as group weighted loss functions [60] and domain classifiers [19] are not immediately extendable to this setup. Thus, ARM methods may be uniquely suited to be paired with domain learning.

## Acknowledgments and Disclosure of Funding

MZ thanks Matt Johnson and Sharad Vikram for helpful discussions and was supported by an NDSEG fellowship. HM is funded by a scholarship from the Dr. Tech. Marcus Wallenberg Foundation for Education in International Industrial Entrepreneurship. AG was supported by an NSF graduate research fellowship. CF is a CIFAR Fellow in the Learning in Machines and Brains program. This research was supported by the DARPA Assured Autonomy and Learning with Less Labels programs.

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
