# A  Broader Impacts

Though machine learning systems have been deployed in many real world domains with great success, data that is anomalous or structurally different from the training data still sometimes renders these systems unreliable, harmful, or even dangerous. It is necessary, in order to realize the full potential of machine learning "in the wild", to have effective methods for detecting, robustifying against, and adapting to distribution shift. The potential upsides of developing such methods are clear. Imagine systems for image classification that fix incorrect or offensive outputs by adapting to each end user, or self driving cars that can smoothly adapt to driving in a new setting. We believe our work is a small step toward the goal of adapting in the face of distribution shift.

However, there are also complications and downsides that must be considered. For example, it is important to understand the failure modes and theoretical limits to handling distribution shift, otherwise we may place "false confidence" in our deployed systems, which may be catastrophic. Our work does not address this aspect of the problem, though this is an important direction for future work. Perhaps more insidiously, this line of research may grant even greater capabilities to parties that are able to collect larger and larger datasets. Deep learning systems are capable of effectively learning from ever growing data, and as the training data grows, the system can be trained to better adapt to a wider range of potential shifts. Thus, it is imperative to continue to push for high quality open source datasets, so that we may democratize the tools of machine learning.

# B  More Details on the ARM Methods

A schematic of the ARM-CML method is presented in Figure 4. The post adaptation model parameters $\theta'$ are $[\theta, \mathbf{c}]$. Since we only ever use the model after adaptation, both during training and at test time, we can simply specify $g(\mathbf{x}; \theta') = f_{\mathrm{pred}}(\mathbf{x}, \mathbf{c}; \theta)$. Though not strictly necessary, we could define the behavior of $g$ before adaptation, i.e., with unadapted parameters $\theta$, as using a running average of the context computed throughout the course of training, similar to BN. We then also see that $h$ is a function that takes in $(\theta, \mathbf{x}_1, \ldots, \mathbf{x}_K)$ and produces $\left[\theta, \frac{1}{K}\sum_{k=1}^{K} f_{\mathrm{cont}}(\mathbf{x}_k; \phi)\right]$. In the streaming setting, we keep track of the average context over the previous test points $\mathbf{c}$ and we maintain a counter $t$ of the number of test points seen so far.[4] When we observe a new point $\mathbf{x}$, we increment the counter and update the average context as $\frac{t}{t+1}\mathbf{c} + \frac{1}{t+1}f_{\mathrm{cont}}(\mathbf{x}; \phi)$, and then we make a prediction on $\mathbf{x}$ using this updated context. Notice that,

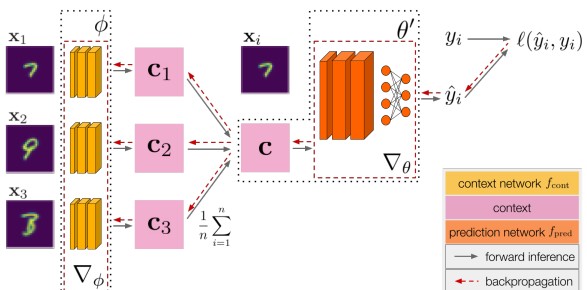

Figure 4: During inference for ARM-CML, the context network produces a vector $\mathbf{c}_k$ for each input image $\mathbf{x}_k$ in the batch, and the average of these vectors is used as the context $\mathbf{c}$ is input to the prediction network. This context may adapt the model by providing helpful information about the underlying test distribution, and this adaptation can aid prediction for difficult or ambiguous examples. During training, we compute the loss of the post adaptation predictions and backpropagate to update $\theta$ and $\phi$.

with this procedure, we do not need to store any test points after they are observed, and this procedure results in an equivalent context to ARM-CML in the batch test setting after observing $K$ data points.

In the streaming setting, ARM-BN is similar to ARM-CML, though slightly more complex due to the requirement of computing second moments. Denote the context after seeing $t$ test points as $\mathbf{c} = [\boldsymbol{\mu}, \boldsymbol{\sigma}^2]$, the mean and variance of the BN layer activations on the points so far. Upon seeing a new test point, let $\mathbf{a}$ denote the BN layer activations computed from this new point, with size $h$. We

---

[4]An alternative to maintaining a counter $t$ is to use an exponential moving average, though we do not experiment with this.

then update $\mathbf{c}$ to be

$$\left[ \frac{ht}{h(t+1)}\boldsymbol{\mu} + \frac{\sum \mathbf{a}}{h(t+1)}, \frac{ht}{h(t+1)}(\boldsymbol{\sigma}^2 + \boldsymbol{\mu}^2) + \frac{\sum \mathbf{a}^2}{h(t+1)} - \left( \frac{ht}{h(t+1)}\boldsymbol{\mu} + \frac{\sum \mathbf{a}}{h(t+1)} \right)^2 \right].$$

Again note that we do not store any test points and that we arrive at the same context as the batch test setting after observing $K$ data points.

Though we did not evaluate ARM-LL in the streaming setting, in principle this method can be extended to this setting by performing a gradient step with a smaller $\alpha$ after observing each test point. In an online fashion, similar to Sun et al. [69], we can continually update the model parameters over the course of testing rather than initializing from the meta-learned parameters for each test point.

## C Contrasting with Prior Benchmarks

One aim of this work is to identify problems for which unlabeled adaptation is feasible, helpful, and potentially crucial, inspired by important real world problem settings such as federated learning. Thus, the problems we focus on will naturally differ from prior work in domain shift, which have different implicit and explicit goals when designing and choosing benchmarks. In particular, as discussed in Section 1, many prior benchmarks assume the existence of a consistent input-output relationship across domains, for which various methods can be designed to try and better uncover this relationship. Compared to problems where adaptation is important, we can roughly characterize these benchmarks as having a conditional distribution $p(y|\mathbf{x})$ that is more stable across domains and thus does not depend as much on the marginal $p(\mathbf{x})$. As Blanchard et al. [8] note informally, and as mentioned in Section 3, the less information the marginal provides about the conditional, the less we expect domain generalization strategies to improve over ERM. Indeed, Gulrajani and Lopez-Paz [23] provide a comprehensive survey of domain generalization benchmarks and find that, though ERM is sometimes outperformed by certain methods on certain benchmarks, ERM is competitive with the state of the art on average across the benchmarks.

Gulrajani and Lopez-Paz [23] also evaluated ARM-CML across the whole suite and found middling performance across most of the testbeds in the benchmark. This negative result provides further evidence that adaptation may not be well suited to these problems, at least in their standard formulations. Similarly, adaptation and ARM methods also do not improve performance on some of the WILDS domain generalization problems [35], potentially due to the marginal $p(\mathbf{x})$ not providing much information about $p(y|\mathbf{x})$, or other factors such as the lack of training domains or shared structure between domains. One potentially interesting result that Gulrajani and Lopez-Paz [23] found was that ARM-CML did outperform ERM and all prior methods on one toy problem: the colored MNIST benchmark [3]. For a non adaptive model, the goal as originally proposed in Arjovsky et al. [3] is to disregard color and learn the invariant relationship between digits and labels. Irrespective of the original motivations, though, an adaptive model is in theory capable of learning a more flexible classification strategy for this problem, in that it may leverage information about the current domain in order to produce better predictions. Viewed this way, it becomes clear why ARM-CML can learn a more performant solution for the colored MNIST problem. This result on a toy problem provides further motivation for identifying and studying real world problems for which adaptation can be beneficial, alongside other benchmarks geared toward discovering invariances.

Specifically when viewing learning adaptation as a meta-learning problem, as in this work, we may pose additional hypotheses about a problem's desired properties. For example, in meta-learning, each task is viewed as a "higher level" data point, and this generally motivates constructing many different tasks so as to prevent the learner from overfitting to the tasks. We extend this intuition to our work in that our problems have tens to hundreds of domains, whereas the benchmarks in DomainBed have between 3 to 6 domains. Note that the overall dataset sizes are still comparable, so previous benchmarks typically also have orders of magnitude more data per domain. Depending on the scenario, it may be difficult to either collect data from many domains, or conversely it may be difficult to collect many data points from any single domain. For example, the FEMNIST dataset naturally contains hundreds of users each contributing at most hundreds of examples, but it would be difficult to collect orders of magnitude more data from any given user. These practical considerations should also factor into the choice of algorithm for solving any particular problem.

Prior testbeds used in group distributionally robust optimization (DRO) typically also contain a small number of groups [60], and these testbeds also have a couple of other important differences. First, as discussed in Section 5, group DRO testbeds typically use the same training and test groups and measure worst case performance, which differs from domain generalization, meta-learning, and most problems considered in this work, which construct or hold out disjoint sets of domains for testing. Second, prior group DRO testbeds use label information to construct groups, in that data within each group will all have the same label. This is not an issue for non adaptive models, however, classification in this setup becomes much easier for adaptive models and particularly if training with an ARM method, as the model simply needs to learn to adapt to output a constant label. Thus, in this work, we identify and set up problems which are distinct from both prior work in domain generalization and group DRO, in order to properly evaluate ARM and prior methods in settings for which adaptation is beneficial.

## D  Additional Experimental Details

Code for Table 1 results will be available from `https://github.com/henrikmarklund/arm`.

In our experiments, we use several different computing clusters with either NVIDIA Titan X Pascal, RTX 2080 Ti, or V100 GPUs, and all experiments use 1 GPU. When reporting our results, we run each method across three seeds and reported the mean and standard error across seeds. Standard error is calculated as the sample standard deviation divided by $\sqrt{3}$. We checkpoint models after every epoch of training, and at test time, we evaluate the checkpoint with the best worst case validation accuracy. Training hyperparameters and details for how we evaluate validation and test accuracy are provided for each experimental domain below. All hyperparameter settings were selected in preliminary experiments using validation accuracy only.

We also provide details for how we constructed the splits for each dataset. These splits were designed without any consideration for the train, validation, and test accuracies of any method. All of these design choices were made either intuitively – such as maintaining the original data splits for MNIST – or randomly – such as which users were selected for which splits in FEMNIST – or with a benign alternate purpose – such as choosing disjoint sets of corruptions with mostly different types.

### D.1  Rotated MNIST details

We construct a training set of 32292 data points using 90% of the original training set – separating out a validation set – by sampling and applying random rotations to each image. The rotations are not dependent on the image or label, but certain rotations are sampled much less frequently. Rotations of 0 through 20 degrees, inclusive, have 7560 data points each, 30 through 50 degrees have 2160 points each, 60 through 80 have 648, 90 through 110 have 324 each, and 120 to 130 have 108 points each.

We train all models for 200 epochs with mini batch sizes of 300. We use Adam updates with learning rate 0.0001. We construct an additional level of mini batching for our method as described in Section 4, such that the batch dimensions of the data mini batches are $6 \times 50$ rather than just 300, and each of the inner mini batches contain examples from the same rotation. We refer to the outer batch dimension as the *meta batch size* and the inner dimension as the batch size. All methods are still trained for the same number of epochs and see the same amount of data. DRNN uses an additional learning rate hyperparameter for their robust loss, which we set to 0.01 across all experiments [60].

We compute validation accuracy every 10 epochs. We estimate validation accuracy on each rotation by randomly sampling 300 of the held out 6000 original training points and applying the specific rotation, resampling for each validation evaluation. This is effectively the same procedure as the test evaluation, which randomly samples 3000 of the 10000 test points and applies a specific rotation.

We retain the original $28 \times 28 \times 1$ image dimensionality, and we divide inputs by 256. We use convolutional neural networks for all methods with varying depths to account for parameter fairness. For all methods that do not use a context network, the network has four convolution layers with 128 $5 \times 5$ filters, followed by $4 \times 4$ average pooling, one fully connected layer of size 200, and a linear output layer. Rectified linear unit (ReLU) nonlinearities are used throughout, and BN [31] is used for the convolution layers. The first two convolution layers use padding, and the last two convolution layers use $2 \times 2$ max pooling. For ARM-CML and the context ablation, we remove the first two convolution layers for the prediction network, but we incorporate a context network. The context

network uses two convolution layers with 64 filters of size $5 \times 5$, with ReLU nonlinearities, BN, and padding, followed by a final convolution layer with $12$ $5 \times 5$ filters with padding.

## D.2 FEMNIST details

FEMNIST, and EMNIST in general, is a significantly more challenging dataset compared to MNIST due to its larger label space (62 compared to 10 classes), label imbalance (almost half of the data points are digits), and inherent ambiguities (e.g., lowercase versus uppercase "o") [12]. In processing the dataset,[5] we filter out users with fewer than 100 examples, leaving 262, 50, and 35 unique users and a total of 62732, 8484, and 8439 data points in the training, validation, and test splits, respectively. The smallest users contain 104, 119, and 140 data points, respectively. We keep all hyperparameters the same as for rotated MNIST, except we set the meta batch size for ARM methods to be 2, and we use stochastic gradient updates with learning rate 0.0001, momentum 0.9, and weight decay 0.0001. For DANN, we use Adam updates with learning rate 0.0001 as stochastic gradient updates were unsuccessful for this method.

We compute validation accuracy every epoch by iterating through the data of each validation user once, and this procedure is the same as test evaluation. Note that all methods will sometimes receive small batch sizes as each user's data size may not be a multiple of 50. Though this may affect ARM methods, we demonstrate in Section 5 that ARM-CML and ARM-BN can adapt using small batch sizes, such as in the streaming test setting. The network architectures are the same as the architectures used for rotated MNIST, except that, when applicable, the last layer of the context network has only 1 filter of size $5 \times 5$.

## D.3 CIFAR-10-C and Tiny ImageNet-C details

For both CIFAR-10-C and Tiny ImageNet-C, we construct training, validation, and test sets with 56, 17, and 22 domains, respectively. Each domain is based on type and severity of corruption. We split domains such that corruptions in the training, validation, and test sets are disjoint. Specifically, the training set consists of Gaussian noise, shot noise, defocus blur, glass blur, zoom blur, snow, frost, brightness, contrast, and pixelate corruptions of all severity levels. Similarly, the validation set consists of speckle noise, Gaussian blur, and saturate corruptions, and the test set consists of impulse noise, motion blur, fog, and elastic transform corruptions of all severity levels. For two corruptions, spatter and JPEG compression, we include lower severities (1-3) in the training set and higher severities (4-5) in the validation and test sets. In this way, we are constructing a more challenging test setting, in which the test domains are not sampled identically as the training domains, since the corruption types are largely different between the two sets. For the training and validation sets, each domain consists of 1000 images for CIFAR-10-C and 2000 images for Tiny ImageNet-C, giving training sets of size 56000 and 112000, respectively. We use the full test set of 10000 images for each domain, giving a total of 220000 test images for both datasets.

In these experiments, we use a support size of 100 and meta batch size of 3. For CIFAR-10-C, we use the same convolutional network architecture as for rotated MNIST and FEMNIST, except for the first layer which needs to be modified to handle RGB images. For Tiny ImageNet-C, we fine tune ResNet-50 [24] models pretrained on ImageNet. The context ablation and ARM-CML additionally use small convolutional context networks, and the learned loss ablation and ARM-LL use small fully connected loss networks. For this domain, we further incorporate BN adaptation into the context ablation and ARM-CML, as we found this technique to generally be very helpful when dealing with image corruptions. The images are first normalized by the ImageNet mean and standard deviation. For CIFAR-10-C, we train models from scratch for 100 epochs, and for Tiny ImageNet-C we fine tune for 50 epochs. We use stochastic gradient updates with learning rate 0.01, momentum 0.9, and weight decay 0.0001. We evaluate validation accuracy after every epoch and perform model selection based on the worst case accuracy over domains. We perform test evaluation by randomly sampling 3000 images from each domain and computing worst case and average accuracy across domains.

---

[5]https://github.com/TalwalkarLab/leaf/tree/master/data/femnist.

Table 3: Comparing to DANN [19] as an unsupervised domain adaptation (UDA) method, in which the particular test domain is known at training time. Note that this involves retraining models for each test evaluation, and ARM-CML is still more performant by leveraging meta-training and adaptation.

| Method | Rotated MNIST | |
|---|---|---|
| | WC | Avg |
| DANN (DG) | $79.7 \pm 1.1$ | $95.0 \pm 0.1$ |
| DANN (UDA) | $82.4 \pm 1.6$ | $94.9 \pm 0.2$ |
| ARM-CML | $\mathbf{88.2 \pm 0.5}$ | $\mathbf{96.5 \pm 0.2}$ |

Table 4: Comparing to a modified version of ARM-CML with probabilistic contexts, similar to Kumagai and Iwata [37]. The standard formulation of ARM-CML performs better on rotated MNIST and FEMNIST, possibly due to the objective purely encouraging predictive accuracy.

| Method | Rotated MNIST | | FEMNIST | |
|---|---|---|---|---|
| | WC | Avg | WC | Avg |
| ARM-CML | $\mathbf{88.2 \pm 0.5}$ | $\mathbf{96.5 \pm 0.2}$ | $\mathbf{71.3 \pm 1.2}$ | $\mathbf{86.4 \pm 0.3}$ |
| ARM-CML w/ prob. $\mathbf{c}$ | $82.6 \pm 0.6$ | $93.8 \pm 0.5$ | $65.1 \pm 2.5$ | $84.7 \pm 0.7$ |

# E    Additional Experiments

## E.1    Additional comparisons

In Table 3 and Table 4, we provide additional comparisons to unsupervised domain adaptation (UDA) methods and zero shot domain adaptation methods, respectively. A number of methods have been proposed for UDA, and for simplicity, we compare to DANN [19], which we evaluated in Section 5 as a domain generalization algorithm but was originally proposed for UDA. When faced with multiple test shifts, UDA methods run training separately for each shift, as they assume access to unlabeled samples from the test distribution at training time. For rotated MNIST, where there are 14 test groups, evaluating DANN as a UDA method involved 42 separate training runs, as we still used 3 training seeds per test evaluation. We see Table 3 that DANN in this setting seems to perform better in terms of worst case accuracy, though the error margins are overlapping, which is not surprising given each model's ability to specialize to a particular test domain. However, by leveraging meta-training and adaptation, ARM-CML still performs the best on this problem.

As noted in Section 2, Kumagai and Iwata [37] propose a method for zero shot domain adaptation that is quite similar to ARM-CML, with the primary high level difference being that, in their method, the contexts are treated as probabilistic latent variables. We thus evaluate a variant of ARM-CML in which we placed a unit Gaussian prior independently on each dimension of the context $\mathbf{c}$ and optimized an evidence lower bound. In Table 4, we see that this variant generally performed worse than the original formulation of ARM-CML, possibly due to the objective balancing between satisfying a restrictive prior and optimizing for predictive accuracy.

## E.2    Additional results with loosened assumptions

In Figure 5, we include results for the rotated MNIST problem in the test streaming setting. We can see the same general trend as for Tiny ImageNet-C, where the models trained via ARM methods are able to adapt successfully, and in this easier domain these models require fewer than 10 test inputs to reach their performances reported in Table 1, where adaptation is performed with batches of 50 points.

In the case of unknown domains, one option is to use unsupervised learning techniques to discover domain structure in the training data. To test this option, we focus on rotated MNIST and ARM-CML, which performs the best on this dataset, and train a variational autoencoder (VAE) [34, 58] with discrete latent variables [32, 47] using the training images and labels.

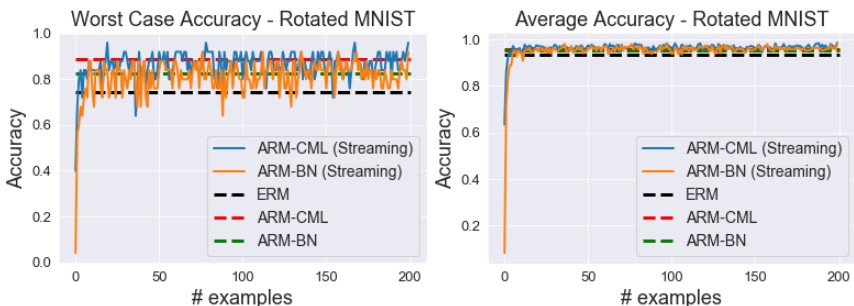

Figure 5: On rotated MNIST, ARM methods reach strong performance in the streaming setting after fewer than 10 data points, again despite meta-training with batch sizes of 50.

Table 5: Using learned domains, ARM-CML outperforms ERM and matches the performance of TTT on rotated MNIST. This result may be improved by techniques for learning more diverse domains.

| Method | WC | Avg |
|---|---|---|
| ERM | $74.3 \pm 1.7$ | $93.6 \pm 0.4$ |
| TTT | $\mathbf{81.1 \pm 0.3}$ | $\mathbf{95.4 \pm 0.1}$ |
| ARM-CML | $\mathbf{81.7 \pm 0.3}$ | $\mathbf{95.2 \pm 0.3}$ |

We define the latent variable, which we denote as $c$ to differentiate from the domain $z$, to be Categorical with 12 possible discrete values, which we purposefully choose to be smaller than the number of rotations. The VAE is not given any information about the ground truth $z$; however, we weakly encode the notion that $c$ is independent of $y$ by conditioning the decoder on the label. We use the VAE inference network to assign domains to the training data, and we run ARM-CML using these learned domains. In Table 5, we see that ARM-CML in this setting outperforms ERM and is competitive with TTT, which as discussed earlier encodes a strong inductive bias, via rotation prediction, for solving this task. Figure 6 visualizes samples from the

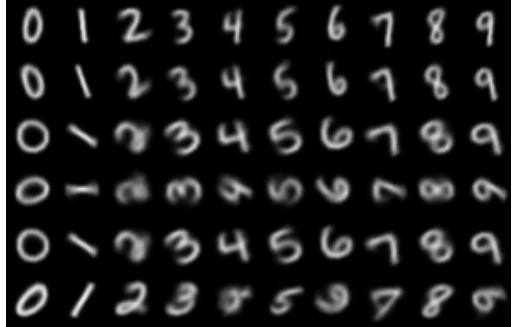

Figure 6: VAE samples conditioned on different values of $y$ (x axis) and $c$ (y axis). The VAE learns to use $c$ to represent rotations.

VAE for different values of $y$ and $c$, which shows that the VAE learns to encode rotation information using $c$. This result suggests that, when domain information is not provided, a viable approach may be to learn domains which then enables the use of ARM methods.