# OpenReview forum: "Adaptive Risk Minimization: Learning to Adapt to Domain Shift"
_NeurIPS.cc/2021/Conference — NeurIPS 2021 Poster_

### Official Review · Reviewer_sJVX · 2021-06-25

**Rating:** 7
**Confidence:** 4

**Summary:**

This paper proposes a framework of adaptive risk minimization (ARM), which adapts to unlabeled instances at the test phase
by pretraining with labeled instances obtained in many groups (domains or users).
To adapt to unlabeled instances, the authors proposed three approaches (ARM-CML, ARM-BN, and ARM-LL).
The effectiveness of the proposed method was shown using real-world image datasets.

**Ethical Concerns:**

No ethical concerns

**Limitations And Societal Impact:**

Yes

**Main Review:**

Pros:
- Problem formulation is interesting.
- Experiments with sequential arriving data are interesting.

Cons:
- Some related works are missing.
- Experiments can be improved.

Details:

This paper proposed an interesting and useful problem setting.
Unlike the standard meta-learning methods that require labeled instances as support set at the test phase,
this approach requires unlabeled instances at the test phase to adapt to new groups.
Although the proposed method for adaptation is straightforward and simple, I like this research direction.
But, I think that there are some improvements that can be made.

First, about the experiment, how does the proposed method compare to the unsupervised domain adaptation,
which require both labeled training data and unlabeled test data during training?
I think that this comparison gives us a better insight into the proposed approach.
In addition, this paper uses only image datasets in the experiments.
If the authors can show the effectiveness of the proposed method on datasets other than image datasets,
it will strengthen the contribution of the paper (although it is not always necessary).

Second, the problem setting of this paper is not completely new.
For example, the paper [1] proposes almost the same problem setting and a method similar to the ARM-CML.
This method uses only unlabeled instances to adapt to new target domains at the test time by learning labeled instances in multiple source domains.
For adaptation, the deepset network, which outputs a latent context vector from the set of unlabeled instances, was used.
I think it is necessary to discuss in more depth the differences between these methods and the advantages of the proposed method.

[1] Kumagai, Atsutoshi, and Tomoharu Iwata. "Zero-shot domain adaptation without domain semantic descriptors." arXiv preprint arXiv:1807.02927 (2018).

===after rebuttal===
Thank you for your answer.
I think the additional experimental comparison will be useful. I would like to raise the score from 6 to 7, provided that the authors reflect these results in the revised version and discuss them in more depth.

**Time Spent Reviewing:**

3

---

> ### Author Response · Authors · 2021-08-10
> **Initial Response to Reviewer sJVX**
>
> Thank you for your helpful comments and suggestions! Here, we will address each of your concerns and describe the changes we will make to the final paper -- these include adding new results to compare to unsupervised domain adaptation and the paper you pointed out.
>
> > “how does the proposed method compare to the unsupervised domain adaptation”
>
> We will add to the final paper results on the rotated MNIST testbed for DANN (reference [16] in the current paper) in the domain adaptation problem setting, where it has knowledge of the test domain at training time. The results can be summarized as follows:
>
> | Method | Rotated MNIST WC | Rotated MNIST Avg |
> | :--- | ---: | ---: |
> | DANN (DG) | 80.3 $\pm$ 1.4 | 95.1 $\pm$ 0.1 |
> | DANN (UDA) | 82.4 $\pm$ 1.6 | 94.9 $\pm$ 0.2 |
> | ARM-CML | **88.7 $\pm$ 0.6** | **96.7 $\pm$ 0.1** |
>
> We can see that, on worst case accuracy, DANN performs better on average in this setting compared to the domain generalization setting, likely because it has privileged information about the test domain. However, ARM methods can still perform better by leveraging unlabeled adaptation at test time. Note that these results involved 42 separate training runs of DANN, as there are 14 different test domains and results were computed with 3 seeds per domain. Thus, unsupervised domain adaptation is more computationally intensive than ARM. We are happy to include any additional details that you would like in a follow-up response.
>
> > “[‘Zero-shot Domain Adaptation without Domain Semantic Descriptors’] proposes almost the same problem setting and a method similar to the ARM-CML”
>
> > “necessary to discuss in more depth the differences between these methods and the advantages of the proposed method”
>
> Thank you for pointing out this relevant paper! Implementation details aside, the primary difference between this method and ARM-CML is that this method interprets the latent context variable to be probabilistic, thus imposing a prior and optimizing an evidence lower bound, whereas ARM-CML is an end-to-end method for learning a non-probabilistic context variable. We will add to the final paper results on rotated MNIST and FEMNIST for this method, which we implemented ourselves as a modification of ARM-CML. The results can be summarized as:
>
> | Method | Rotated MNIST WC | Rotated MNIST Avg | FEMNIST WC | FEMNIST Avg |
> | :--- | ---: | ---: | ---: | ---: |
> | ARM-CML | **88.7 $\pm$ 0.6** | **96.7 $\pm$ 0.1** | **67.8 $\pm$ 1.3** | **85.7 $\pm$ 0.3** |
> | ARM-CML w/ prob. context | 82.6 $\pm$ 0.6 | 93.8 $\pm$ 0.5 | **65.1 $\pm$ 2.5** | 84.7 $\pm$ 0.7 |
>
> Generally, we found that the performance of this method was slightly worse than ARM-CML for this problem, possibly due to the objective trading off between predictive accuracy and regularization of the latent context variable. We are happy to include any additional details that you would like in a follow-up response. We will also add an in-depth discussion of this prior work to the related work section, and we will conduct an additional literature review to add other zero-shot domain adaptation papers that we may have missed.
>
> Again, thank you for helping to improve the paper! We believe we have addressed all of your points, but please let us know if you have any additional questions or concerns.

---

> > ### Comment · Reviewer_sJVX · 2021-08-22
> > **Thank you for your response**
> >
> > Thank you for your answer.
> > I think the experimental comparison between these existing methods will be useful. I would like to raise the score from 6 to 7, provided that the authors reflect these results in the revised version and discuss them in more depth.

---

### Official Review · Reviewer_dgsw · 2021-07-09

**Rating:** 6
**Confidence:** 4

**Summary:**

This work proposes to modify MAML where the inner optimization problem is adapting to a different distribution in an unsupervised manner rather than finetuning on a new task. That is, the inner step must reduce loss but without the labels.

**Limitations And Societal Impact:**

Sure

**Main Review:**

I find this work to be quite limited in novelty, and it seems like a very shallow application. Given the similarities between domain generalization, domain adaptation, and meta-learning/multi-task learning, it's not insightful to just apply the MAML objective, and so the contribution of this work seems to be trying out a few types of adaptation to find which ones work reasonably well.

In fact, I think this work is better described as "MAML when the test task samples are unlabeled" rather than a method for distribution shift. The proposed algorithm is _literally exactly MAML_, where the inner gradient steps are replaced with alternative adaptation techniques for when the labels are unavailable. **The fact that this work is basically just "how can we slightly modify MAML" is I think the primary factor for me, but below I include some additional discussion which is also relevant and I think essential to properly address.**

I feel the discussion of general "distribution shift" is poorly framed. While Blanchard et al. did indeed consider a prior over tasks, the current formulation of domain generalization is a much more difficult problem of generalizing in a _minimax_ sense, when the statistics across domains may have very little in common (further, note that that work provides actual theoretical guarantees, where this work provides none).

This paper heavily conflates "group shift" with general distribution shift, such as in the problem of domain generalization. In the title and abstract, the authors indicate that this method is meant for dealing with group shift---this is not the same thing as the typical domain generalization setup or distribution shift in general. This is a _very_ specific type of change at test time where the mixture components of a set of fixed sub-populations (all of which are observed at train time) changes. That's fine, but I think the introduction discussing domain generalization and general distribution shift is quite misleading because this paper addresses a very niche problem that's nowhere near as difficult as the problems discussed in the introduction. The experiments show that the algorithm is successful at unsupervised meta-learning, or adapting to new domains which share a _great deal of structure_ with the training data. The experiments do _not_ show that the algorithm is successful at generalization under general distribution shift.

As further evidence, and particularly damning, I note that in Appendix C the authors discuss the performance of ARM on the DomainBed benchmark. See the results here:
https://github.com/facebookresearch/DomainBed/blob/master/domainbed/results/2020_10_06_7df6f06/results.png
I find it quite disingenuous that the authors write: "ARM-CML handily outperforms ERM and all prior methods on the colored MNIST benchmark", as CMNIST is an extremely contrived and unrealistic dataset. Aside from the two MNIST-based datasets (the two easiest ones), on _all other datasets in the DomainBed benchmark_ ARM is nowhere near the top, and on average it is quite in the middle of the pack, failing to outperform ERM. **Cherry-picking results on a benchmark suite defeats the purpose, and gives me the impression that this comparison was made in bad faith.**

I think "MAML when the test task labels are unavailable" is an ok idea, lacking in novelty but not totally without merit. I also personally feel that group shift is not a realistic assumption for what to expect at test time without some a priori structural knowledge of the task we are trying to solve. Regardless, the experimental results demonstrate its ability to perform well when there is heavily shared structure among the domains (as might be expected, given the empirical success of MAML). But the algorithm does not work for more difficult distribution shift (and there is *no theoretical intuition or justification whatsoever*), so really this is just a slight tweak to an existing approach to meta-learning.

This paper would greatly benefit from an extensive reframing, and the results---positive _and_ negative---need to be honestly and frankly reported.

--------------------
**Update after rebuttal**. The authors have indicated that they will fix the issues that I surfaced, an after reconsideration I consider this to be marginally above the publication threshold.

I think it is **essential** that these changes are actually implemented. This method works well, but only in *very* specific settings, with *much* stronger assumptions than previously made. As an example, the comparison in table 1 stratifies by assumption, but then still only bolds the top performing algorithm. This gives the casual reader the impression that ARM achieves SOTA, when in reality it is simply not being compared to any other algorithms with the same access to information at test time. This fact should be conveyed **much more clearly**, in addition to the other issues I found with the experimental reporting and general discussion of domain generalization and group shift.

**In fact, I think the algorithm suggested here is more akin to unsupervised domain adaptation than domain generalization**... maybe the comparisons should be updated to reflect this.

**Time Spent Reviewing:**

4

---

> ### Author Response · Authors · 2021-08-10
> **Initial Response to Reviewer dgsw**
>
> Thank you for your helpful and detailed comments and suggestions! Here, we will address each of your concerns and describe the changes we will make to the final paper -- these changes include scoping the problem setting more clearly in the introduction, adding new results on the WILDS benchmark [1], carefully revising Appendix C, clarifying our contributions in the introduction and method sections, and providing relevant theory based on prior work.
>
> > “I feel the discussion of general ‘distribution shift’ is poorly framed.”
>
> > “This paper heavily conflates ‘group shift’ with general distribution shift, such as in the problem of domain generalization.”
>
> > “The experiments do not show that the algorithm is successful at generalization under general distribution shift.”
>
> In the final paper, we will use the term “domain” in place of “group”, and we will revise the introduction and experiments sections to clarify that our focus is on the domain generalization setting. Specifically, we evaluate on only one subpopulation shift problem: rotated MNIST, where all groups are observed during training and the primary challenge is group imbalance. The other three problems are domain generalization problems, as the goal is to predict on new unseen users or corruptions at test time. We agree that these testbeds are less extreme or “difficult” than the DomainBed testbeds, in which domains “may have very little in common”. But as the DomainBed paper shows, these existing testbeds are very difficult, to the point that just improving in-distribution generalization via a well-tuned ERM implementation is competitive with state-of-the-art domain generalization methods. We will clarify in the introduction that our goal is specifically to tackle domain generalization problems where the new domains “share a great deal of structure” with seen domains. We believe that there is considerable value in studying these problems, e.g., for predicting on new users, cameras, hospitals, or experimental batches as in some of the WILDS datasets [1].
>
> We believe that the changes detailed above will clarify the scope of this paper and make it more apparent why ARM methods should be expected to work well in this specific setting. Please let us know if these proposed changes would address your concerns regarding scope.
>
> > “group shift is not a realistic assumption for what to expect at test time without some a priori structural knowledge of the task”
>
> We agree, which is why we feel it is important to clarify that we primarily focus on domain generalization. We believe that domain generalization accurately describes many problems of practical interest for which “a priori structural knowledge of the task” is available and can be used as leverage for obtaining better solutions. As additional evidence for this, we will add to the final paper new results on the WILDS benchmark [1], a suite of distribution shift problems specifically designed to be tractable while faithfully representing important real world applications. We have so far evaluated on three domain generalization problems from WILDS -- iWildCam, Camelyon17, and RxRx1 -- and the current results can be briefly summarized as:
>
> | Method | iWildCam Test Acc | Camelyon17 Test Acc | RxRx1 Test Acc |
> | :--- | ---: | ---: | ---: |
> | ERM | **71.6 $\pm$ 2.5** | 70.3 $\pm$ 6.4 | 29.9 $\pm$ 0.4 |
> | CORAL | **73.3 $\pm$ 4.3** | 59.5 $\pm$ 7.7 | 28.4 $\pm$ 0.3 |
> | BN adaptation | 46.6 $\pm$ 0.7 | **88.6 $\pm$ 1.0** | 20.0 $\pm$ 0.1 |
> | ARM-BN | **70.3 $\pm$ 1.9** | **87.2 $\pm$ 0.7** | **31.2 $\pm$ 0.1** |
>
> From this, we can see that different strategies work best for different distribution shift problems. CORAL, a method for invariance, improves accuracy on iWildCam the most on average, though the error margins overlap; BN adaptation in general greatly improves Camelyon17 results, though without meta-training it significantly harms performance on the other two problems; and ARM-BN is competitive on all three problems and the most effective method for RxRx1. We will include full results, both positive and negative, on all of the WILDS datasets in the final paper and are happy to include any additional details that you would like in a follow-up response.
>
> > “I find [Appendix C] quite disingenuous”
>
> > “Cherry-picking results on a benchmark suite defeats the purpose, and gives me the impression that this comparison was made in bad faith.”
>
> Upon second look, we agree that the exposition in Appendix C can be misleading, and we will revise it as follows. First, we will explicitly note that the DomainBed authors evaluated ARM-CML and found middling performance (note that we did not run these experiments, the DomainBed authors did). Second, we will clarify that colored MNIST is a toy dataset.
>
> Our intent is not to hide or cherry pick any results. We do not claim that better performance on colored MNIST indicates a better practical method, nor do we wish to imply that the negative results on the rest of DomainBed are insignificant. We will make an extensive proofreading pass of Appendix C in order to better convey our intended point: that certain datasets (including some real world datasets such as in WILDS) exhibit greater leverage for generalizing to new test domains, while other domain generalization testbeds are very difficult, as noted in our response above. In the final paper, we will ensure that both the positive and negative results on this benchmark are presented appropriately and proportionally.
>
> > “it's not insightful to just apply the MAML objective”
>
> > “I think this work is better described as ‘MAML when the test task samples are unlabeled’”
>
> > “this work is basically just ‘how can we slightly modify MAML’”
>
> In the final paper, we will revise the introduction and method sections to address these points.
> In fact, we agree that our paper is a straightforward extension of meta-learning (though not just MAML, as we note in the next paragraph). But we would like to clarify that applying meta-learning to learn adaptive models for domain generalization problems is the main contribution of this paper, rather than devising new methods for meta-learning. We would further emphasize that, to the best of our knowledge, this is the first paper to actually propose this idea and demonstrate its effectiveness, and we see the simplicity of this idea as a strength.
>
> We note that the meta-learning objective, which is indeed similar to Eq (1), predates MAML (see, e.g., [2] below) and has been subsequently used in a number of works (see, e.g., [3] below). We believe it is no more fair to criticize this paper in this regard than it is to criticize any of these papers, including MAML, for reusing this objective. The novelty of this paper and all of these prior works is not in devising a new objective, but rather in coming up with new meta-learning models and methods, proposing new problem settings that are of practical interest, and demonstrating improved performance.
>
> > “[Blanchard et al] provides actual theoretical guarantees, where this work provides none”
>
> > “there is no theoretical intuition or justification whatsoever”
>
> We will revise the preliminaries section to expand the discussion of Blanchard et al (reference [6] in the current paper) to discuss how their theoretical foundation applies to ARM methods as well. In particular, Lemma 9 in [6] provides justification for the ARM problem setting and objective, and to make this clear, we provide this prior result as it will appear in the final paper (with additional introductory text for completeness) here: https://drive.google.com/file/d/1rrg07lxzctqwwkZxjaLvHwLjuVGjlsz5
>
> Again, thank you for helping to improve the paper! We believe we have addressed all of your points, but please let us know if you have any additional questions or concerns.
>
> [1] Koh et al, “WILDS: A Benchmark of in-the-Wild Distribution Shifts”. ICML 2021.
>
> [2] Ravi and Larochelle, “Optimization as a Model for Few-Shot Learning”. ICLR 2017.
>
> [3] Requeima et al, “Fast and Flexible Multi-Task Classification Using Conditional Neural Adaptive Processes”. NeurIPS 2019.

---

> > ### Comment · Reviewer_dgsw · 2021-08-13
> > **Conditioned on these improvements, I've reconsidered.**
> >
> > I'm glad to hear the under-reporting of experimental results was not intentional. Just to wrap up in response to these points:
> >
> > **We note that the meta-learning objective...subsequently used in a number of works.** This is true, but I think of MAML as the paper which just said "hey let's use ERM and gradient descent on this 'task-prior' idea" and found that it worked really well. I am essentially using the word "MAML" to reference that general concept in meta-learning.
> >
> > **The novelty of this paper...and demonstrating improved performance.** Here I think is where my opinion differs from that of the authors. I don't think "meta-learning for X" is typically enough novelty to warrant a paper, unless a real additional problem is solved in the process of learning to modify it for the new setting. I have many times had the thought "hey I bet you could use meta-learning for that". The reason I don't run experiments and write a paper about it is because I think *everyone* has that thought.
> >
> > If this paper demonstrated that adapting meta-learning for this setting is actually quite difficult, and took considerable effort and care to derive, I would in general be much more inclined to recommend publication. But in fact, the section of the paper describing the adaptations is 1 short paragraph per item; it seems essentially like the first couple ideas that were tried (or maybe some more were tried and the ones that worked were selected). This is what I meant by "the contribution of this work seems to be trying out a few types of adaptation to find which ones work reasonably well."
> >
> > On the other hand, I don't expect complexity for complexity's sake, and the results do indicate that this works well for some specific settings, with the caveat that it requires **much** stronger assumptions (and therefore the comparison is not really fair). In particular, I think the table reporting results doesn't emphasize this enough. **For example, only the top scorer in each column is bolded, when in reality the bolded comparisons should be stratified by algorithm assumptions. This gives the casual reader the impression that ARM achieves SOTA, when in reality it is simply not being compared to any other algorithms with the same access to information at test time**. In particular, a fair comparison would be to include ARM *without* any sort of streaming or batching in the same group as the other algorithms (I realize this would not make sense for ARM, my point is just that these algorithms are meant for fundamentally different settings).
> >
> > I also think that general domain generalization is extremely difficult, and lacking in meaningful assumptions which make it more feasible; I like the "streaming" setting and think it's worth exploring further. So, I think with appropriate reporting of results and proper discussion of the strong conditions under which this method works, I would consider this paper to be marginally above the acceptance threshold.
> >
> > I am upgrading my recommendation and I sincerely hope the authors will appropriately edit the paper as they have indicated.

---

> > > ### Author Response · Authors · 2021-08-16
> > > **Thank you for your response!**
> > >
> > > Thank you for your prompt reply!
> > >
> > > > “only the top scorer in each column is bolded, when in reality the bolded comparisons should be stratified by algorithm assumptions”
> > >
> > > We will correct Table 1 in the paper such that each stratification contains its own bolded results, and we will further clarify these differences in assumptions across methods when discussing the results in the experiments section. For completeness, here is an anonymous link showing Table 1 as it will appear in the final paper: https://drive.google.com/file/d/1jlpHEoSxlRo9dn0pTNOwh-KDbTHbzVx6.
> > >
> > > We will ensure that all of the changes detailed in our initial response will appear as stated in the final paper. Please let us know if you have any additional questions or concerns, we are open to continuing this useful conversation!

---

### Official Review · Reviewer_6LKK · 2021-07-14

**Rating:** 7
**Confidence:** 3

**Summary:**

This paper proposes a framework for unsupervised multi-source domain adaptation in which training samples are grouped according to which domain they come from. (The paper uses "groups" to refer to the domains, but I will simply use "domains".) The authors assume that predictions can be adapted using the information of the input distribution of the target domain, and the learner is allowed to use a batch of unlabeled test samples at test time in order to adapt predictions.

The authors approach this problem by learning an adaptation model h that inputs a batch of unlabeled samples and transforms the parameters of a prediction model g (Algorithm 1). They present three concrete methods based on this approach and compare them with previous methods using a few benchmark datasets, which confirms the effectiveness of the approach.

**Limitations And Societal Impact:**

The paper does not mention societal impact although I have no suggestions on that.

**Main Review:**

# Overall evaluation
The clarity of the paper could be improved, but the core idea of the proposed approach is interesting and the experimental results look promising. I recommend accepting the paper provided that the authors address clarity issues.

# Originality
* As far as I understand, the problem setup can be viewed as multi-source unsupervised domain adaptation, and this itself is not new.
* The proposed approach of learning to adapt using a batch of unlabeled samples seems novel as far as I can see after seeing several papers.

# Quality
* The paper provides good discussions on related work.
* The technical part is clear except for the concrete algorithms.
* The general algorithm is natural and reasonable although the paper does not provide any theoretical guarantees.
* On the other hand, the concrete algorithms look quite heuristic and it would be nice if there were theoretical analysis.
* The experiments are well-designed and the results look convincing.

# Clarity
* Section 1 and Section 2 are somewhat too abstract and unclear to me. In fact, I could not understand the problem setup until I read Section 4. A reason may be the authors try to avoid technical terminology and give high-level arguments in the introduction part, which had unfortunately negative impact on my reading experience.
* For example, the authors use "groups", which is too general and ambiguous, for describing what I would call "domains".
* Also, the introduction does not seem to explain well what the proposed method really does.
* I suggest defining the problem setup a little bit more formally in early part of the paper so that readers can have clear understandings what the paper is talking about.
* The descriptions of the concrete methods are hard to understand without looking at the supplementary material.

# Significance
* The problem setup is slightly more restrictive than the standard unsupervised domain adaptation in that we need samples from multiple source-domains (groups).
* However, there may be many real-world applications indeed satisfying the requirement such as the examples that the authors present in the paper.
* The idea of incorporating information of the input distribution into adaptation is interesting and seems reasonable. It does put strong assumptions like covariate shift but seems to be solving more general problems.
* The experimental results look convincing and indeed show superiority of the proposed method.

# Other comments
* It would be nice to add discussions about previous work on multi-source unsupervised domain adaptation and zero-shot domain adaptation.


-----

# Update after the authors' response
I have read the other review comments and the authors' response. My opinion has not changed, and I would like to keep my score at 7. I hope the authors improve the clarity of the paper and the experiment part, as discussed in the rebuttal.

**Time Spent Reviewing:**

8 hours

---

> ### Author Response · Authors · 2021-08-10
> **Initial Response to Reviewer 6LKK**
>
> Thank you for your helpful and detailed comments and suggestions! Here, we will address each of your concerns and describe the changes we will make to the final paper -- these changes include scoping the problem setting more clearly in the introduction, expanding the description of each ARM method in Section 5.2, providing relevant theory in the preliminaries section, and expanding the related works section.
>
> > “Section 1 and Section 2 are somewhat too abstract and unclear to me.”
>
> > “For example, the authors use ‘groups’, which is too general and ambiguous, for describing what I would call ‘domains’.”
>
> > “Also, the introduction does not seem to explain well what the proposed method really does.”
>
> Thank you for the detailed suggestions for improving the exposition. In the final paper, we will use the term “domain” in place of “group” and emphasize in the introduction that our problem setting is most closely related to domain generalization. This will allow us to bring in concrete terminology and notation much earlier, which will improve the clarity of the introduction and allow for more detailed explanation of the proposed approach. This change will also naturally propagate into Section 2, thus also allowing for a more clear and detailed explanation of the illustrative examples.
>
> > “The technical part is clear except for the concrete algorithms.”
>
> > “The descriptions of the concrete methods are hard to understand without looking at the supplementary material.”
>
> To address your comments, we will significantly expand the description of each proposed ARM method, primarily by moving the important details in Appendix B to Section 5.2. To make room, we will shorten the experiments section and move some discussion of the results to Appendix E.
>
> > “the paper does not provide any theoretical guarantees”
>
> We will revise the preliminaries section to expand the discussion of Blanchard et al (reference [6] in the current paper) to discuss how their theoretical foundation applies to ARM methods as well. In particular, Lemma 9 in [6] provides justification for the ARM problem setting and objective, and to make this clear, we provide this prior result as it will appear in the final paper (with additional introductory text for completeness) here: https://drive.google.com/file/d/1rrg07lxzctqwwkZxjaLvHwLjuVGjlsz5
>
> > “It would be nice to add discussions about previous work on multi-source unsupervised domain adaptation and zero-shot domain adaptation.”
>
> We will expand the related works section in the final paper to include additional references and discussion for these two areas as well as domain generalization, as these terms are sometimes used interchangeably. If you have any specific references that are important, we would be happy to include and discuss them.
>
> > “The paper does not mention societal impact”
>
> We do discuss this in Appendix A, but did not have room for it in the main paper. In the final paper, we will move parts of Appendix A to the discussion section in order to better highlight what we believe are the potential societal impacts of this work.
>
> Again, thank you for helping to improve the paper! We believe we have addressed all of your points, but please let us know if you have any additional questions or concerns.

---

### Official Review · Reviewer_dMsf · 2021-07-16

**Rating:** 6
**Confidence:** 4

**Summary:**

The authors propose a method to adapt to group distribution shifts using meta-learning. Their proposed method is general for adapting to different test distributions. Empirical results show that their method is effective in improving the classification accuracies of common datasets by 1-4%.


**Limitations And Societal Impact:**

Yes

**Main Review:**

- The key observation of the authors is that changes in the underlying group distributions in input x can tell us something about the distribution of labels, and this is a natural problem that occurs for example with the different handwriting styles of different users in handwriting recognition.

- I think the authors can provide more details on the different adaptation approaches in ARM-CML, ARM-BN, and ARM-LL, especially ARM-LL. This is an important part of the algorithm and crucial for understanding why it works, but only half a page is used for these methods.

- When I look at Algorithm 1, apart from the sampling of z from different groups, there is nothing specific in the algorithm that is tailored for group distribution shifts. It can be applied to other types of test distribution shifts as well, as long as we have a sample of the test data. This is both a pro and a con: it is general enough to apply to other distribution shifts, but on the other hand there does not seem to be a lot of new insights and contributions to the base meta-learning approach.

- For the empirical evaluations, the results are in general good and contain small improvements over previous adaptation approaches.

- Overall this is a borderline paper to me because although the problem of adapting to group distribution shifts is interesting, the current paper does not add a lot on top of a direct application of the meta-learning method.



**Time Spent Reviewing:**

3

---

> ### Author Response · Authors · 2021-08-10
> **Initial Response to Reviewer dMsf**
>
> Thank you for your helpful comments and suggestions! Here, we will address each of your concerns and describe the changes we will make to the final paper -- these changes include clarifying our contributions in the introduction and method sections, adding new results on the WILDS benchmark [1], and expanding the description of each ARM method in Section 5.2.
>
> > “there does not seem to be a lot of new insights and contributions to the base meta-learning approach”
>
> > “the current paper does not add a lot on top of a direct application of the meta-learning method”
>
> In the final paper, we will revise the introduction and method sections to address these points. In fact, we agree that our paper is a "direct application of [meta-learning]". But we would like to clarify that applying meta-learning to learn adaptive models for handling test shifts is the main contribution of this paper, rather than devising new methods for meta-learning. We would further emphasize that, to the best of our knowledge, this is the first paper to actually propose this idea and demonstrate its effectiveness, and we see the simplicity of this idea as a strength.
>
> In handling test distribution shifts, such as new domains or image corruptions, the primary approaches have been methods for robustness and invariance. On the other hand, meta-learning has generally not been considered as a tool for these problems. We aim to demonstrate that meta-learning can be successfully applied to tackle test shifts, and we propose the ARM framework for doing so. This framework allows us to easily extend existing meta-learning paradigms to learning unlabeled adaptation strategies, and we show that the resulting ARM methods can be effective for these problems.
>
> To further this point, we will add to the final paper new results on the WILDS benchmark [1], a suite of distribution shift problems specifically designed to be tractable while faithfully representing important real world applications. We have so far evaluated on three domain generalization problems from WILDS -- iWildCam, Camelyon17, and RxRx1 -- and the current results can be briefly summarized as:
>
> | Method | iWildCam Test Acc | Camelyon17 Test Acc | RxRx1 Test Acc |
> | :--- | ---: | ---: | ---: |
> | ERM | **71.6 $\pm$ 2.5** | 70.3 $\pm$ 6.4 | 29.9 $\pm$ 0.4 |
> | CORAL | **73.3 $\pm$ 4.3** | 59.5 $\pm$ 7.7 | 28.4 $\pm$ 0.3 |
> | BN adaptation | 46.6 $\pm$ 0.7 | **88.6 $\pm$ 1.0** | 20.0 $\pm$ 0.1 |
> | ARM-BN | **70.3 $\pm$ 1.9** | **87.2 $\pm$ 0.7** | **31.2 $\pm$ 0.1** |
>
> From this, we can see that different strategies work best for different distribution shift problems. CORAL, a method for invariance, improves accuracy on iWildCam the most on average, though the error margins overlap; BN adaptation in general greatly improves Camelyon17 results, though without meta-training it significantly harms performance on the other two problems; and ARM-BN is competitive on all three problems and the most effective method for RxRx1. We will include full results on all of the WILDS datasets in the final paper and are happy to include any additional details that you would like in a follow-up response.
>
> > “I think the authors can provide more details on the different adaptation approaches”
>
> To address your comments, we will significantly expand the description of each proposed ARM method, primarily by moving the important details in Appendix B to Section 5.2. To make room, we will shorten the experiments section and move some discussion of the results to Appendix E.
>
> Again, thank you for helping to improve the paper! We believe we have addressed all of your points, but please let us know if you have any additional questions or concerns.
>
> [1] Koh et al, “WILDS: A Benchmark of in-the-Wild Distribution Shifts”. ICML 2021.

---

> ### Author Response · Authors · 2021-08-17
> **Following up with Reviewer dMsf**
>
> We are following up to see whether our initial response has addressed all of your points, or if you have additional questions or concerns? Specifically, the final paper will:
>
> 1. move details from Appendix B to Section 5.2, to allow for more in-depth explanations of each ARM method,
> 2. clarify that the main contribution of this paper is to demonstrate that meta-learning tools can be readily adapted to tackle domain generalization problems, and
> 3. present new results on the WILDS benchmark to further support this main contribution.
>
> Please let us know if you have other points; we are eager to continue this conversation in order to further improve the paper.

---

> > ### Comment · Reviewer_dMsf · 2021-08-26
> > **Thank you for the response**
> >
> > Thank you for the authors' response. I appreciate the addition of the explanations of each ARM method, and extra experiments. I still maintain my opinion that the approach does not add a lot on top of meta-learning, but I am happy to increase my score to 6.

---

### Decision · Program_Chairs · 2021-09-27

**Decision:**

Accept (Poster)

**Comment:**

This paper proposes to apply a meta-learning method (similar to MAML) to solve the domain generalization problem. After spending some time reading the paper and trying to understand the main idea, I agree with Reviewer dMsf, sJVX, and dgsw that there is a limited novelty in terms of both methodology and theory. Firstly, from the domain generalization perspective, the idea of learning models that can adapt to a collection of unlabeled data at test time has appeared as early as in [1, 2] and subsequent works in this direction. In these works, the models also take as input the empirical marginal distributions through the kernel function on them. Hence, the key distinction here is the use of deep learning trained in a meta-learning style as a way to equip the models with adaptation capability. Secondly, from the meta-learning perspective, the proposed idea is very similar to the classic MAML, as also pointed out by Reviewer dgsw. The key difference from the standard setting is the use of unlabeled data alone. Lastly, the authors seem to miss several important works in both domain generalization and meta-learning when discussing the related works in Section 3. I encourage the authors to properly conduct a literature review as this is an increasingly important area of research. In the final version, I encourage the authors to improve the clarity and to better highlight the key contributions of this work.

Nevertheless, there is a positive consensus among the expert reviewers that the experimental results are promising, and that the rebuttal has adequately addressed their concerns. The authors are willing to incorporate the reviewers' suggestions in improving the final version of this work. Hence, I recommend acceptance for publication at NeurIPS2021 as a poster.

Remark: Line 75-76: "Invariance methods would fare no better, as there is no feature space that can separate the ambiguous points and make them classifiable". This is indeed misleading. One can in fact separate the ambiguous points if the group/domain information is incorporated into the feature representation as has been commonly done in the literature.

- [1] G. Blanchard, G. Lee, and C. Scott. Generalizing from several related classification tasks to a new unlabeled sample. In Advances in Neural Information Processing Systems 24, pages 2178–2186, 2011.

- [2] Muandet, K., Balduzzi, D., Schölkopf, B., Domain generalization via invariant feature representation. In: Proceedings of the 30th International Conference on Machine Learning (ICML13). pages 10–18, 2013.